# Cell-type-specific mRNA transcription and degradation kinetics in zebrafish embryogenesis from metabolically labeled single-cell RNA-seq

Lior Fishman [1], Avani Modak [2], Gal Nechooshtan [1], Talya Razin[1], Florian Erhard [3,4], Aviv Regev [5,6], Jeffrey A. Farrell [2] ✉ & Michal Rabani [1] ✉

During embryonic development, pluripotent cells assume specialized identities by adopting particular gene expression profiles. However, systematically dissecting the relative contributions of mRNA transcription and degradation to shaping those profiles remains challenging, especially within embryos with diverse cellular identities. Here, we combine single-cell RNA-Seq and metabolic labeling to capture temporal cellular transcriptomes of zebrafish embryos where newly-transcribed (zygotic) and pre-existing (maternal) mRNA can be distinguished. We introduce kinetic models to quantify mRNA transcription and degradation rates within individual cell types during their specification. These models reveal highly varied regulatory rates across thousands of genes, coordinated transcription and destruction rates for many transcripts, and link differences in degradation to specific sequence elements. They also identify cell-type-specific differences in degradation, namely selective retention of maternal transcripts within primordial germ cells and enveloping layer cells, two of the earliest specified cell types. Our study provides a quantitative approach to study mRNA regulation during a dynamic spatio-temporal response.

During development, transcript levels are tightly regulated by the combined action of mRNA transcription and degradation. By changing one or more of these processes, cells produce complex expression patterns that allow undifferentiated embryonic cells to establish distinct cell identities within the embryo. While the role of mRNA transcription in these events is very well established[1,2], growing evidence highlights the importance of its precise interplay with mRNA degradation in shaping developmental gene expression patterns. For example, destruction of Nodal agonist and antagonist mRNAs triggered by the zebrafish microRNA miR-430 generates the precise expression levels needed for correct patterning of embryos[3]. In addition, the localized stabilization of maternally inherited mRNAs within zebrafish primordial germ cells initiates a unique gene expression program[4,5], that is only later supplemented by new zygotic transcription[6]. However, despite its critical role in determining mRNA levels, the coordinated regulation of mRNA transcription and degradation remains less studied. It is still unclear to what extent mRNA stability contributes to shaping developmental expression patterns,

[1]Department of Genetics, Silberman Institute of Life Sciences, The Hebrew University of Jerusalem, Edmond J. Safra Campus, Jerusalem 9190401, Israel. [2]Division of Developmental Biology, Eunice Kennedy Shriver National Institute of Child Health and Human Development, NIH, Bethesda, MD 20814, USA. [3]Institute for Virology and Immunobiology, University of Würzburg, Würzburg, Germany. [4]Chair of Computational Immunology, University of Regensburg, Regensburg, Germany. [5]Department of Biology, MIT, Cambridge, MA 02139, USA. [6]Klarman Cell Observatory Broad Institute of MIT and Harvard Cambridge, Cambridge, MA 02142, USA. ✉e-mail: Jeffrey.farrell@nih.gov; michal.rabani@mail.huji.ac.il

and how its coordination with mRNA transcription affects it. It is also unknown if and how regulation changes between genes and cell types of a developing embryo undergoing cell type specification.

Technical and computational challenges have limited the availability of genome-wide data on mRNA transcription and degradation during development. Conventional RNA-Seq[7–9], and more recently also single-cell RNA-Seq (scRNA-Seq)[10–12] measure overall mRNA levels, but provide only a partial view of mRNA dynamics. These approaches cannot distinguish between simultaneous transcription and degradation of transcripts in embryos. To address this challenge, RNA-Seq was combined with strategies for transcriptional arrest[13], quantification of intron-containing pre-mRNAs[13,14], SNP detection[15] or RNA metabolic labeling[16–18]. These strategies have generally used bulk sequencing, obscuring differences between individual cell types and diverse cellular identities within embryos. Recently, metabolic labeling of RNA was also combined with scRNA-Seq within cell culture contexts[19–24] and embryos[25], revealing cellular states and transitions at unprecedented resolution. However, such snapshot experiments are restricted in their ability to dynamically monitor changes in expression level and infer regulatory rates. Application of kinetic modeling tools has greatly enhanced single-cell studies[19,24] and allowed for also monitoring and quantifying the underlying mRNA production and degradation rates in vitro. However, it is still unclear if such tools could be used to study whole embryos with diverse cellular identities and quantitatively separate the contribution of mRNA transcription and degradation in shaping spatio-temporal embryonic gene expression patterns.

The maternal-to-zygotic transition provides a compelling model to dissect the contributions of mRNA transcription and degradation to cell specification. At the onset of development, embryos are transcriptionally silent and rely on maternally-provided mRNAs and proteins that were deposited in the egg[26]. In zebrafish embryos, this maternally controlled period lasts for ~3.3 h, and involves many rounds of cell division. Genome activation starts as early as 2.3 h with a minor wave of transcription. Once the maternal-to-zygotic transition begins in earnest, embryos undergo a massive degradation of maternally inherited mRNAs and begin to produce new zygotic transcripts[26–28]. As the maternal-to-zygotic transition unfolds, cell-type-specific changes in mRNA levels program pluripotent embryonic cells to assume more specialized identities[2]. While transcriptional regulation is critical for these transitions[2], it has remained unclear to what extent regulation of mRNA degradation plays a role in these cell type specification events. For instance, mRNAs for several lineage specific developmental regulators are also maternally deposited (e.g., *tbx16*, *cdx1b*, and others), and thus are initially ubiquitous. To direct downstream patterning, they must achieve a proper cell-type-specific expression, which requires the destruction of non-specific maternal messages. However, how the interplay of mRNA transcription and degradation directs cell-type-specific gene expression patterns within an embryo is still not resolved.

Here, we systematically quantify RNA transcription and degradation rates at cell-type resolution within a developing organism. We combine scRNA-Seq with RNA metabolic labeling within zebrafish embryos, and use them to follow the dynamics of the early zebrafish embryonic transcriptome. We decompose gene expression within single cells into its newly-transcribed (zygotic) and pre-existing (maternal) mRNA components, and develop dynamic models that allow us to resolve the relative contributions of mRNA transcription and degradation in determining spatio-temporal embryonic gene expression programs.

## Results

### Quantifying old and new mRNA in live, single embryonic cells

We combined scRNA-Seq with RNA metabolic labeling to monitor newly-transcribed mRNA accumulation within single cells during zebrafish development (Fig. 1A, Methods). We injected zebrafish embryos

at the one-cell stage with 4sU-triphosphate (4sUTP), which is selectively incorporated into newly-transcribed RNA molecules and distinguishes them from pre-existing, maternally deposited, copies (Methods). To benchmark our approach, we also injected into all embryos in vitro transcribed GFP and mCherry mRNAs that do not include any labeled residues (Methods). As was previously reported[17], injections did not interfere with normal zebrafish development phenotypically or transcriptionally. Injected embryos developed normally and reached the expected developmental stages. Single-cell transcriptomes obtained from these embryos are comparable to published[11] scRNA-Seq datasets of untreated zebrafish embryos (Supplementary Fig. 1A).

To detect the incorporation of label within live cells, we adapted the Drop-Seq[11,29] method by adding a chemical conversion of 4sU residues[30] after mRNA capture on beads (Methods). This conversion alters base pairing during reverse transcription and creates characteristic T-to-C changes in downstream sequencing reads, allowing quantification of newly-transcribed labeled RNAs. Post-treatment transcriptomes were comparable to controls that were not subjected to the additional conversion step for both native genes (Supplementary Fig. 1B-C) and injected controls (Supplementary Fig. 1D). These indicate that conversion did not interfere with the integrity of single-cell transcriptomes generated by Drop-Seq and did not increase barcode mixing between cells.

Using this approach, we collected 8226 single-cell transcriptomes of live cells from embryos that were metabolically labeled starting at the one-cell stage (Fig. 1A). We profiled embryos at three developmental stages following the onset of zygotic transcription (at 3.3 hpf): dome (4.3 hpf, 1855 cells), 30% epiboly (4.8 hpf, 3052 cells from 2 replicates) and 50% epiboly (5.3 hpf, 3319 cells from 2 replicates). Indeed, the frequency of T-to-C conversion increased over developmental stages (Fig. 1B), as expected when more newly-transcribed mRNAs have accumulated in cells.

During these developmental stages, embryos activate a massive degradation of maternally deposited mRNAs and simultaneously initiate new zygotic transcription. As labeling is applied immediately after fertilization, labeled nucleotides should only incorporate into newly-transcribed zygotic mRNAs, while pre-existing maternal transcripts that were produced prior to label injection should not show any significant labeling signal. To show the specificity of metabolic labeling to zygotically transcribed mRNAs, we analyzed a subset of known zygotic genes, and found high T-to-C conversion rates at all stages, as expected. On the other hand, conversion rates of both known maternal genes and injected controls remained low, at similar levels to untreated samples (Supplementary Fig. 2A).

We applied GRAND-SLAM analysis[21] to determine the fraction of newly-transcribed zygotic mRNA from T-to-C conversions for each gene in each cell. This approach deduces the fraction of labeled mRNA per gene from characteristically low 4sU incorporation rates (estimated 5.5% to 8.5% for zygotic genes in our samples, Supplementary Fig. 2A). It uses statistical inference to integrate the underlying position-specific incorporation rates, genetic polymorphism and other confounding effects, and improves accuracy of the estimated labeled fractions compared to the raw T-to-C conversion signals. Indeed, GRAND-SLAM analysis correctly estimated (Supplementary Fig. 2B) low labeling of pre-existing injected controls (labeled fraction <0.8%, averaged across all cells) and known maternal genes (labeled fractions <3.5%); as well as high labeling of newly-transcribed known zygotic genes (labeled fractions >80%). The accuracy of estimated zygotic mRNA fractions was further improved by applying GRAND-SLAM to aggregated pseudo-bulk samples (Supplementary Fig. 2C), reaching nearly 100% for known zygotic genes (Supplementary Fig. 2D).

As expected, estimated labeled fractions within cells increased over developmental stages as more newly-transcribed zygotic mRNAs have accumulated in cells (Fig. 1C). Labeled zygotic mRNAs accounted

on average for only 13% of cellular mRNAs in early dome stage (4.3 hpf, Fig. 1C), increasing to an average of 41% in late 50% epiboly stage (5.3 hpf). These results were also comparable to estimates by bulk RNA-Seq of metabolically labeled embryos. In bulk samples we detected only minimal levels of zygotic mRNAs in 1 hpf samples (4-cell, 0.3% in polyA+, 3.4% in ribo-depleted, Fig. 1D). However, the fraction of zygotic mRNAs increased substantially by 4.8 hpf (30% epiboly, 17.6% in polyA+, 26% in Ribo-depleted), and was similar to estimates in our single-cell analysis (31% on average at 4.8 hpf sample, Fig. 1C).

Taken together, these results validate our method's ability to accurately monitor newly-transcribed zygotic mRNAs and distinguish them from pre-existing maternal copies of genes in single cells of whole embryos. Our method provides a powerful and accurate approach that combines droplet-based scRNA-Seq and metabolic labeling to separately measure levels of maternally provided and zygotically transcribed mRNA within individual cells of a developing organism.

### Pseudotime analysis of the maternal-to-zygotic transition

To interpret single-cell transcriptomes, we used URD[11] to perform dimensionality reduction, UMAP projection, and clustering without considering whether mRNAs were maternal or zygotic. This analysis partitioned cells into 15 clusters (Supplementary Fig. 3A) reflecting both developmental stage and cell type. Clusters were annotated based on their expression of known cell-type-specific genes and visualized on a UMAP projection (Fig. 2A-C). The UMAP represented well both the known specification events that have occurred at these stages of development (Fig. 2A), and the temporal ordering of the 3 sampled timepoints (Fig. 2B).

The levels of maternally deposited or zygotically transcribed mRNAs across different cells on the UMAP projection successfully recapitulated known developmental expression patterns. For example, zygotic transcription of cell-type-specific zygotic regulators (Fig. 2D) was identified in the mesoderm (e.g., *tbxta* and *gsc*), the ectoderm (e.g., *sox3* and *cxcr4a*), and in the enveloping layer (e.g., *krt8* and *spaca4l*). On the other hand, we recovered only pre-existing copies of maternally provided genes such as *h1m* and *wee2* (Fig. 2E). Additionally, known maternal mRNAs that encode regulators of primordial germ cells (such as *nanos3* and *dnd1)* are appropriately restricted to this cell type (Fig. 2F).

Since specification of cell types in the blastula happens asynchronously, we used URD to compute a developmental pseudotime for each cell, which reflects each cell's transcriptional difference from an early, unspecified cell population (Fig. 2C). This calculation considered only total mRNA levels within cells. When considered separately, the decline of maternal and increase of zygotic mRNAs within cells were tightly associated with developmental pseudotime calculated on total mRNA. Differences in zygotic mRNA fractions between cells explained 70% of pseudotime differences (Pearson $r^2 = 0.7$, Supplementary Fig. 3B). The early undifferentiated cells had few zygotically transcribed messages, whereas cells with later pseudotime had higher fractions of zygotically transcribed messages and corresponded to specified cell types (Supplementary Fig. 3C). Cells in the enveloping layer had the highest zygotic mRNA fractions and pseudotimes (Supplementary Fig. 3D), possibly reflecting the very early specification of this cell type. These results demonstrate that developmental pseudotime accurately captures the accumulation of zygotic transcripts during development.

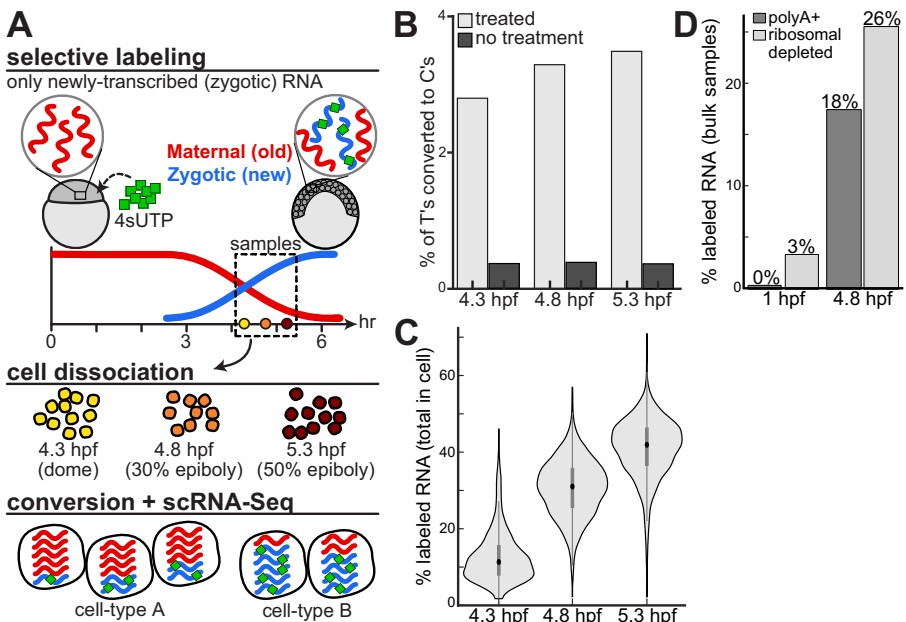

**Fig. 1 | Monitoring the embryonic maternal and zygotic transcriptomes at a single cell resolution. A** An approach to combine scRNA-Seq with RNA metabolic labeling in zebrafish embryos. We injected zebrafish embryos at the one-cell stage with 4sU-triphosphate (4sUTP, green), which is selectively incorporated into newly-transcribed zygotic mRNA molecules (blue), while preexisting maternally contributed molecules (red) remain unlabeled. We collected embryos at three developmental stages following the onset of zygotic transcription: dome (4.3 hpf; yellow), 30% epiboly (4.8 hpf; orange) and 50% epiboly (5.3 hpf; dark red). We dissociated embryos into single cells, and measured their transcriptomes by an adapted scRNA-Seq workflow that included a chemical conversion of labeled residues. Conversion induced T-to-C changes in downstream sequencing reads, enabling the separate quantification of newly-transcribed (zygotic) and pre-existing

(maternal) mRNA within single-cell transcriptomes. **B** Fraction of T bases that were sequenced as C (y-axis) across all genes in the transcriptome, within each of three temporal samples (x-axis), when applying chemical conversion (light gray) or without such treatment (dark gray). **C** Distribution of GRAND-SLAM estimates of the overall percent of labeled RNA within each cell (y-axis) at three developmental stages (x-axis). The central dot is median; gray box bounds are 25th and 75th percentiles, upper and lower limits of whiskers are 1.5x interquartile ranges. Values outside of the upper and lower limits are defined as outliers, $n = 1855$, 3052, and 3319 cells per stage, respectively. **D** GRAND-SLAM estimates of percent of total labeled RNA per sample (y-axis) in bulk samples collected at two developmental stages (1 hpf, 4 cell, and 4.8 hpf, 30% epiboly) using either polyA selection (dark gray) or ribosomal depletion (light gray).

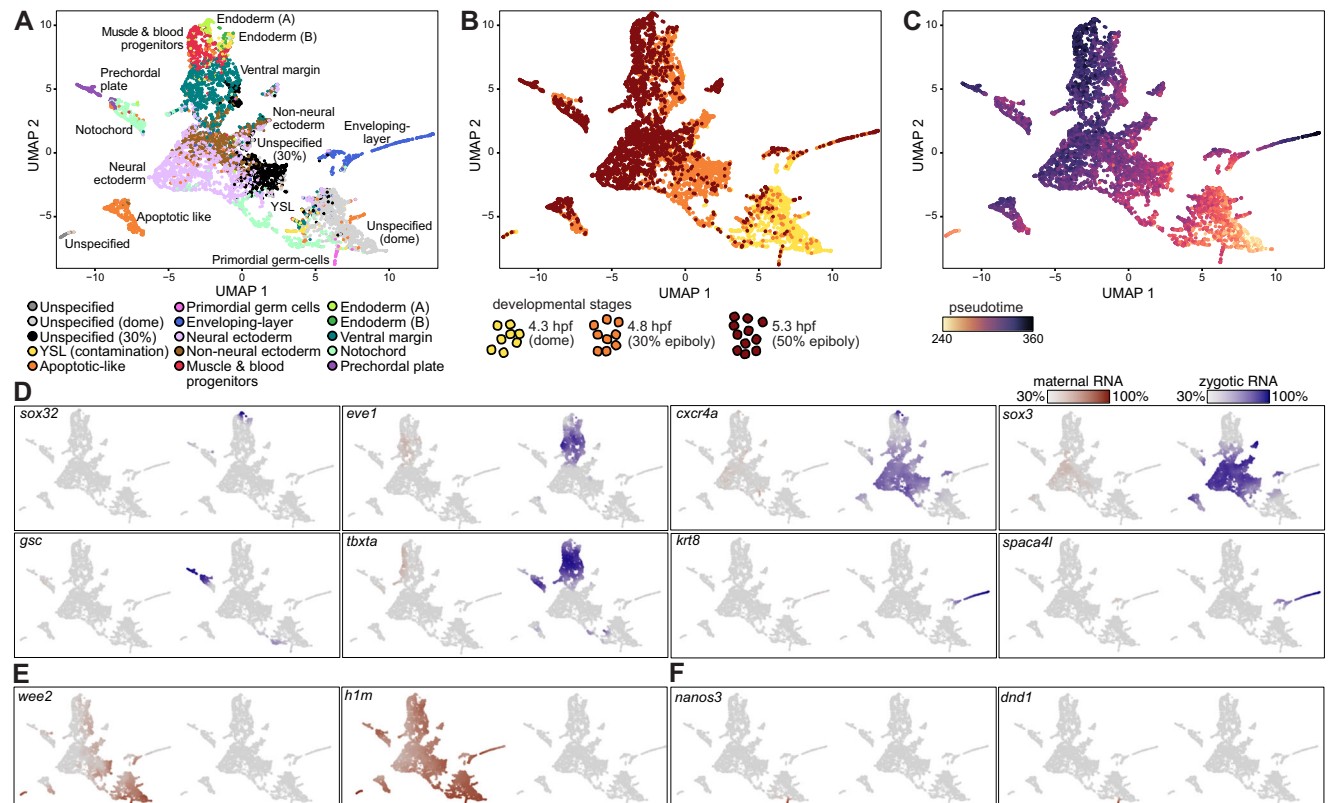

**Fig. 2 | Distinguishing maternal and zygotic expression across single cells of zebrafish embryos.** UMAP projection of 8226 single cells from five embryonic samples (4.3 hpf, 1855 cells; 4.8 hpf, 3052 cells from 2 replicates; 5.3 hpf, 3319 cells from 2 replicates). **A** Colored by 13 distinct cell-type clusters. Three groups of unspecified cells do not express any specific cell-type markers, and represent cells which have still not differentiated. Two of these groups mostly include cells from one specific developmental stage, as indicated. Cells labeled as YSL are likely a contamination that originated from the yolk syncytial layer. Unspecified cells are either from 30% epiboly or dome stages. **B** Colored by developmental stage: 4.3 hpf (dome, 1855 cells, yellow); 4.8 hpf (30% epiboly, 3052 cells, 2 replicates, orange); 5.3 hpf, (50% epiboly, 3319 cells, 2 replicates, dark red). **C** Colored by pseudotime. Values range from 240 pseudo-min (earliest, yellow) to 360 pseudo-min (latest, purple). Pseudotime was calculated using the URD algorithm, based on the cells'

transcriptomic differences from the early unspecified cell populations. **D**–**F** Single-cell expression of developmental genes across all 8226 collected single cells. All cells are plotted, and each cell is colored by the normalized expression of a gene's pre-existing maternal copies (red, left map) or newly-transcribed zygotic copies (blue, right map). The characteristically low label incorporation rates in combination with low per-cell number of reads by scRNA-Seq, limited the accuracy of estimated labeled mRNA fraction within single cells, which was often lower than expected within single cells and resulted in an unlabeled mRNA background. Therefore, we use a gene-specific color scale, scaled by its maximal total expression, and its minimal 30% quantile of maternal and zygotic mRNA expression. Analyzed genes are indicated on plot. **D** Zygotically expressed genes, known to be specifically expressed within a certain lineage. **E** Maternally inherited genes. **F** Maternally inherited genes, known to be specifically expressed within germ cells.

## Most genes are maternally and zygotically contributed

Leveraging the ability of metabolic labeling to quantitatively distinguish the fraction of maternally contributed and zygotically transcribed copies of each gene, we analyzed the average fraction of newly transcribed zygotic mRNAs of each gene across single cells. We partitioned all 6,180 genes into 10 equally sized bins (quantiles) based on their average fraction of newly-transcribed zygotic mRNA in cells (Fig. 3A).

We considered genes within the lowest three quantiles of newly-transcribed mRNA (<5%) as maternal-only transcripts that are not zygotically transcribed during the first 5.3 h of development. These included pluripotency and oocyte regulatory factors (e.g., *h1m*, *wee2*, Fig. 2E), as well as known germ-cell regulators (e.g., *nanos3*, *dnd*, Fig. 2F). This set is also specifically enriched for developmental processes involved in reproduction ($p < 4*10^{-7}$, hypergeometric Set Counts and Sizes (SCS) correction <5%), for endomembrane system ($p < 1*10^{-28}$) and for small molecule metabolism ($p < 2*10^{-12}$).

We considered genes in the top two quantiles of newly-transcribed mRNA (>65%) as zygotic-only transcripts, with a minimal maternally inherited contribution. These included many known zygotic developmental regulators (e.g., *gsc*, *cxcr4a*, Fig. 2D) with documented functions within specific cell lineages. These zygotic-only

transcripts were also enriched for transcription factors ($p < 2*10^{-19}$), patterning ($p < 9*10^{-21}$) and embryo development ($p < 4*10^{-20}$) annotations.

The estimated average fraction of newly-transcribed zygotic copies for the remaining 3090 genes ranged between 5% and 65% (e.g., 47% in *ccnb1*; 8% in *ddx39ab*; Fig. 3B), suggesting they are both maternally contributed and zygotically expressed during the first 5.3 h post-fertilization. Genes in this group were enriched for functions related to RNA metabolism, such as splicing ($p < 1*10^{-40}$), ribosome biogenesis ($p < 5*10^{-31}$) and pol-II transcription ($p < 2*10^{-31}$), as well as other key cellular processes, such as translation ($p < 9*10^{-48}$), cell-cycle ($p < 4*10^{-29}$) and DNA replication ($p < 5*10^{-17}$). Components of these basic functions are both maternally contributed and expressed zygotically immediately following genome activation. Our data shows differences in both rates of maternal mRNA destruction and zygotic mRNA accumulation between transcripts in this group (Fig. 3B). This suggests that both regulatory processes could serve to control expression of these genes during cell type specification. However, in the absence of properly distinguishing maternal and zygotic transcripts as we do in this work, these dynamics would be obscured within total mRNA levels for most genes in zebrafish embryos, which have both maternal and zygotic contributions (Fig. 3A).

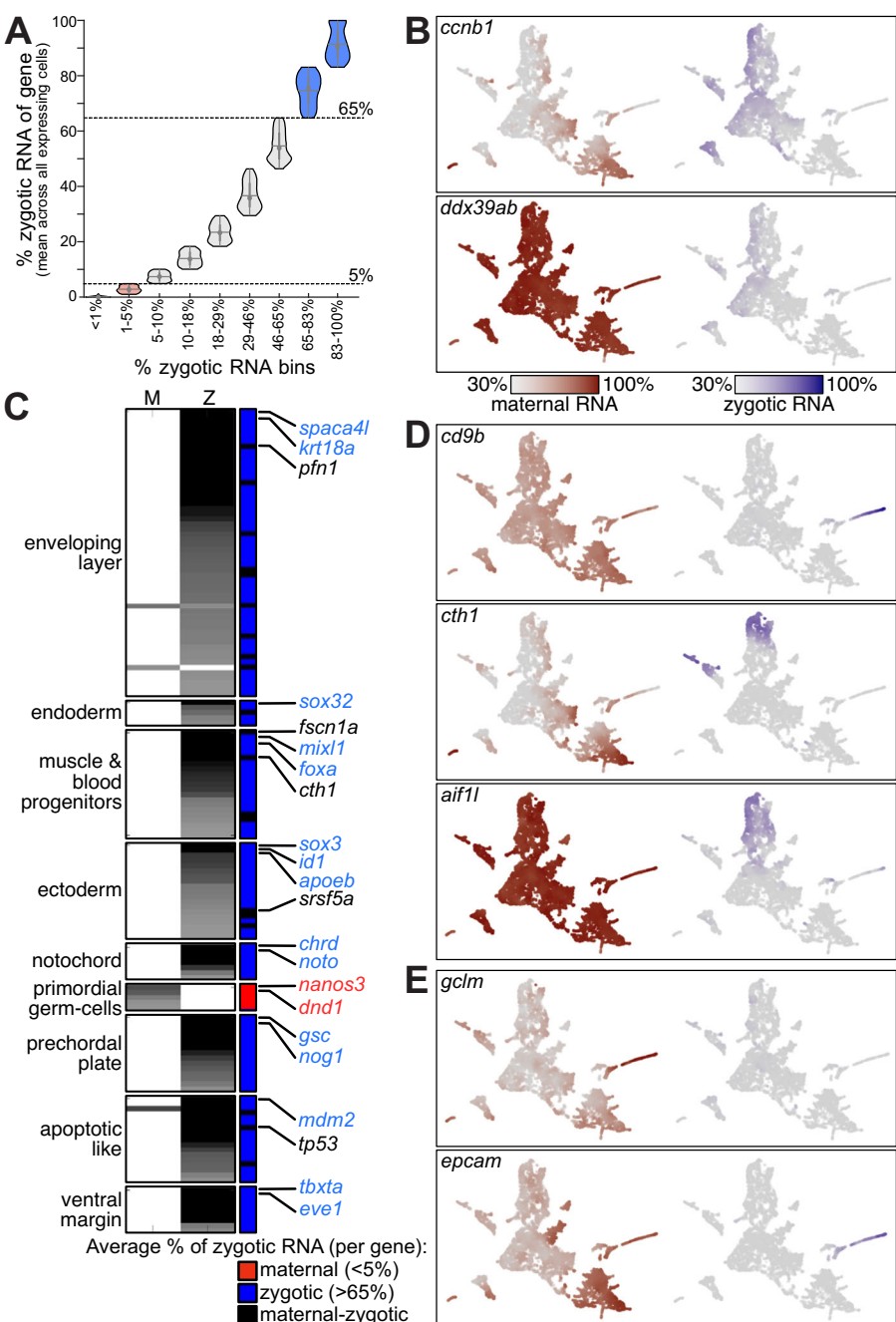

**Fig. 3 | Differences in zygotic mRNA accumulation between genes and cell types. A** Mean percent of zygotic mRNA per gene across all cells that express it with 3 or more UMI counts (y-axis). Genes were divided into 10 bins with an equal number of genes (x-axis). Since the fraction of zygotic mRNA per gene was equal (0%) for transcripts in the lowest two bins, they are plotted as a single set, which is double in size. The central dot is the median; gray box bounds are 25th and 75th percentiles, upper and lower limits of whiskers are 1.5x interquartile ranges. Values outside of the upper and lower limits are defined as outliers, $n = 618$ genes per bin. We consider genes in the top two bins as zygotic (blue, zygotic mRNA fraction >65%) and genes in the bottom three bins as maternal (red, zygotic mRNA fraction <5%, first 2 bins are plotted as a single bin). **B** Expression of genes across 8226 collected single cells. All cells are plotted, and each cell is colored by the normalized expression of a gene's pre-existing (maternal) copies (red, left map) or newly-

transcribed (zygotic) copies (blue, right map). Color scale is gene-specific, and scaled by its maximal total expression and its minimal 30% quantile of maternal and zygotic expression. **C** A set of 154 genes (rows) with either zygotically or maternally restricted expression within cell-types. Grayscale represents *p* values for maternal (left column) or zygotic (middle column) mRNA enrichment within cell-type (indicated on left). Right column is colored by % zygotic mRNA (red: <5%, maternal, 5 genes; blue: >65%, zygotic, 129 genes; black: maternal-zygotic, 20 genes). Maternal-zygotic genes can be cell-type restricted in only one of the two components of their expression (e.g., *cth1* and *pfn1* are only cell-type restricted by their zygotic mRNA, while their maternal mRNA is not restricted). **D** Single cell expression (as in **B**) of maternal-zygotic genes with cell-type-restricted zygotic expression. **E** Single-cell expression (as in **B**) of genes with cell-type-restricted maternal expression in enveloping layer cells.

## Selective RNA stabilization augments zygotic transcription

During the maternal-to-zygotic transition, cell-type-specific expression patterns emerge for many genes. Cell-type-specific expression levels could be achieved in two ways at the maternal-to-zygotic transition:

genes could be differentially transcribed in different cell types, or ubiquitous maternally deposited mRNA could be differentially degraded in different cell types. To investigate the prevalence of these different modes of regulation, we tested each gene for the enrichment of

either its zygotic (newly transcribed) or maternal (pre-existing) expression within different cell types (Methods). Overall, 436 of 6180 genes (7%, Supplementary Data 1) had either zygotically or maternally restricted expression within one or a few cell types (Kolmogorov-Smirnov Bonferroni <1%). Due to the high background in single-cell decomposition of maternal and zygotic mRNA, we conservatively limited further analysis to the 154 of those genes with the highest significance for cell-type specificity (Kolmogorov-Smirnov $p < 10^{-20}$, Fig. 3C).

Though most genes that are expressed in the embryo were both provided maternally and expressed zygotically, genes with cell-type-restricted expression were overwhelmingly zygotic-only (129 genes, 84%, Supplementary Fig. 4A), including most well-known marker genes, such as *tbxta*, *noto* and *eve1* in the mesoderm, *sox17* and *her5* in the endoderm, and *krt5* and *krt8* in the enveloping layer. A much smaller number of maternal-only genes (5 genes, 3%, Fig. 3C) are restricted to a specific cell type. All 5 of these maternal-only genes are restricted to the primordial germ cells and are well-established germ-cell markers[31] (e.g., *dnd1*, *nanos3*, Fig. 2F). We did not identify any additional zygotic markers within this cell type at those stages, consistent with prior findings that these cells are transcriptionally quiescent until later in development. Of these 5 markers, *dnd1*, *nanos3*, and *ddx4* have been shown to be selectively stabilized within zebrafish primordial germ cells[4,5], which is likely the case for the other 2 genes as well (*gra* and *ca15b*). This suggests that selective stabilization of maternal mRNAs is the main mechanism that shapes the primordial germ cell transcriptome during blastula stages, including factors that are crucial for germ cell development.

Finally, we also identified a group of maternal-zygotic genes (20 genes, 13%, Fig. 3C) with restricted expression within cell types. In most cases, only the zygotic copies of these genes were restricted to specific cell types (e.g., *cth1* and *aif1l* in muscle and blood, or *cd9b* in enveloping layer, Fig. 3D), while their maternal copies were not restricted. This indicates that their cell-type-specific patterns are shaped largely by zygotic transcription rather than by cell-type-specific stabilization of maternal mRNA. In these cases, the cell-type-specific expression pattern of the zygotic copies is initially obscured by ubiquitously distributed maternal copies. Some genes exhibit quick elimination of maternal mRNA so that their cell-type-specific expression pattern becomes clearly evident even during blastula stages (e.g., *cth1*, *cd9b*, Fig. 3D, Supplementary Fig. 4B, C). Conversely, some genes exhibit slow degradation of maternal mRNA (e.g., *aif1l*, Fig. 3D, Supplementary Fig. 4D), such that their cell-type-specific transcription is obscured by remaining maternal copies of the same gene until later in development. Notably, two genes with both maternal and zygotic contributions exhibited cell-type-specific enrichment of their maternally contributed copies—the enveloping layer-specific genes, *gclm* and *epcam* (Fig. 3E). This suggests that these genes may achieve enveloping layer-specific expression during blastula stages through cell-type-specific stabilization of maternal mRNAs.

Overall, this analysis supports the expectation that zygotic transcription is the main source of cell-type-restricted expression during the specification of most cell types in zebrafish embryos. However, it also highlights the unique role of maternally inherited mechanisms in establishing the germ cell identity in zebrafish, and suggests that this mode of regulation also occurs within the enveloping layer for some genes. It is notable that the two cell types that exhibit a regulated stabilization of maternal mRNAs are the two earliest specification events during zebrafish embryogenesis[11]. This aligns with the intriguing possibility that zygotic regulation is too late to achieve patterning of these cell types sufficiently early in development.

## Kinetic models quantify gene expression regulatory rates

We aimed to obtain a more quantitative view of the contributions of regulation of maternal and zygotic transcripts to overall mRNA levels during cell type specification, by modeling the dynamics of maternal and zygotic transcriptomes in our data (Fig. 4A). To do so, we built kinetic models which leverage the following characteristics of our data: (1) it measures quantitative gene expression values at single-cell resolution, (2) it clearly distinguishes maternal and zygotic transcripts from the same gene, and (3) it contains multiple time points during early development. These kinetic models effectively allow parameterization of the rates of destruction of maternal mRNA and accumulation of zygotic mRNA and facilitate quantitative comparisons between different genes and different cell types. These rates can be used to reveal regulatory functions that govern gene expression changes in embryos and generate a quantitative view of coordinated transcriptional and posttranscriptional events.

First, we interpolated high resolution expression dynamics from scRNA-Seq data. Since specification of cell types in the blastula happens asynchronously, we used pseudotime (Fig. 2C) to enable more detailed investigation of dynamics than could be achieved with three timepoints. We partitioned the cells into 11 pseudotime bins and estimated the zygotic (newly transcribed) and maternal (preexisting) expression level of 5101 genes within each temporal bin by pseudo-bulk analysis of all cells within a bin (Methods). Gene expression levels within bins were highly correlated to those calculated within three developmental stages (Supplementary Fig. 5A). Moreover, the high-resolution profiles recapitulated the expected expression dynamics of key genes. For example, we recovered upregulation of newly-transcribed mRNA of known zygotic regulators (Fig. 4B), and downregulation of preexisting mRNAs of known maternally provided genes (Fig. 4C). This demonstrates the validity of our approach for interpolating the data between timepoints and calculating higher temporal resolution profiles.

We next developed a kinetic modeling approach to infer separate regulatory rates for maternal and zygotic copies of each gene. We estimated degradation pseudo-rates of maternal copies and accumulation pseudo-rates of zygotic copies per gene. Since our temporal information is based on a pseudotime analysis, the resulting pseudo-rates provide information on the relative production and degradation rates between genes in our dataset, rather than absolute measurements. We used a generative model (Methods, Fig. 4A), in which levels of maternal mRNA are determined by an exponential decay of pre-existing copies (first-order reaction), and those of zygotic transcripts are determined by a linear accumulation of new copies (zero order reaction), assuming minimal degradation. An alternative model which also includes a term for degradation of zygotic mRNAs could significantly improve the fit of only a small fraction of genes (261 genes, 9%, e.g., *cct6a* and *pum1*, Supplementary Fig. 5B), supporting a minimal effect of degradation of zygotic copies on overall mRNA levels within the timeframe of our experiment. We also assumed that, after onset, transcription and degradation rates are constant within the experimental timeframe (spanning less than 2 h of development), but incorporated a gene-specific time of onset for each rate.

We applied maximum-likelihood estimation (Methods) to find the parameters of mRNA production and degradation profiles that best fit the dynamic observations and validated it using simulation studies (Supplementary Fig. 6A, B). Using a "goodness of fit" test (Methods), the fitted kinetic models were retained for 97% of genes (4923 genes at chi-square $p > 0.05$, Supplementary Fig. 6C). As evidence that the determined rates are meaningful, the estimated pseudo-rates correlated to fold-changes measured in bulk SLAM-Seq samples (Pearson correlation > 0.43, Supplementary Fig. 6D). Additionally, the predictions from this kinetic model explained >96% ($R^2$-squared) of the variability in new RNA, old RNA and total RNA expression levels in our data, suggesting that these models with minimal parameters are sufficiently accurate to capture the dynamics of gene expression in the early embryo. Altogether, these establish the ability of our kinetic models to infer kinetic rates of mRNA transcription and degradation

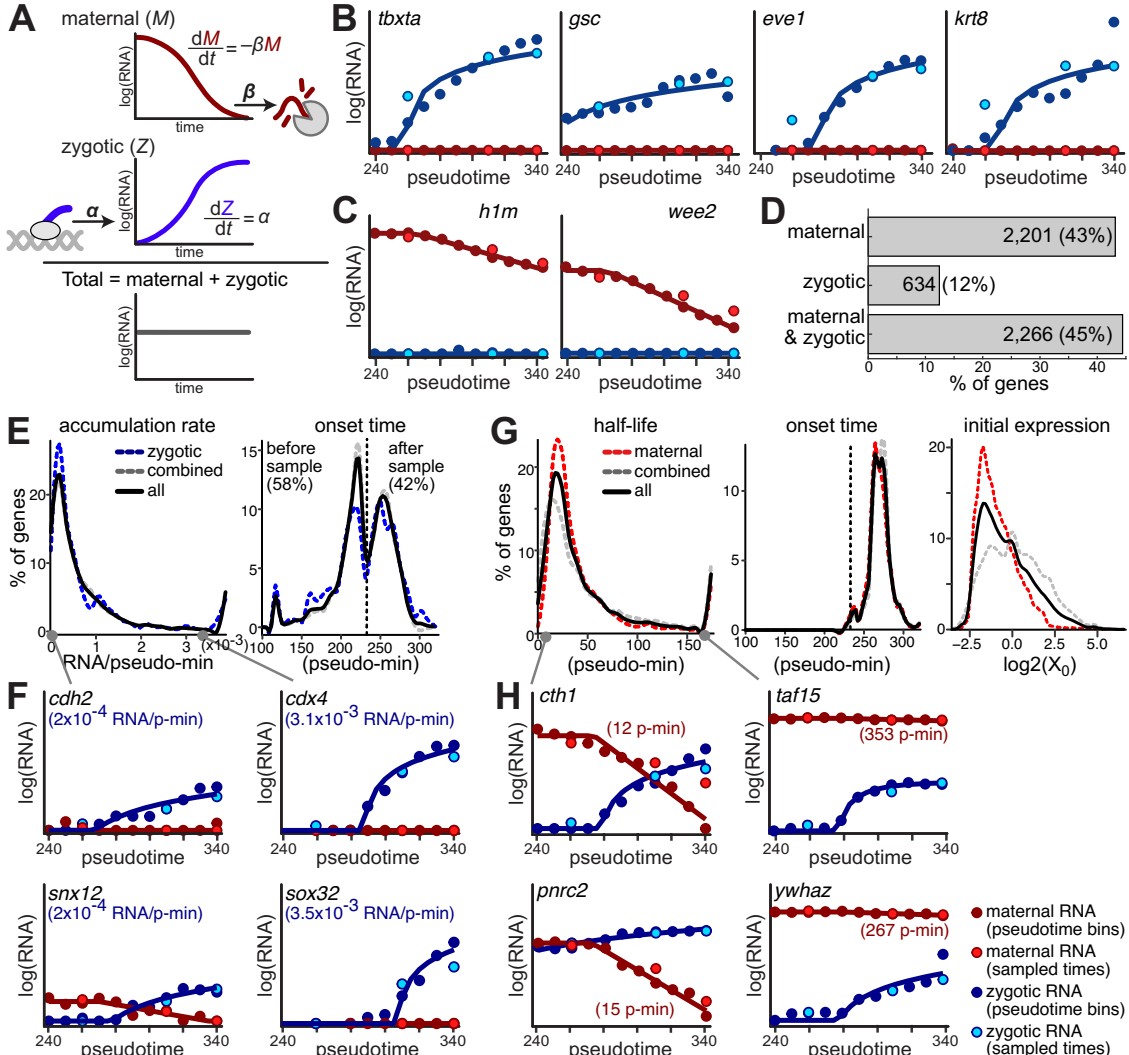

**Fig. 4 | Maternal and zygotic mRNA regulatory kinetic rates shape their temporal expression patterns. A** A kinetic modeling approach to infer per-gene maternal and zygotic kinetic rates. Levels of maternal mRNA (M, red) are determined by an exponential decay of preexisting copies with a constant degradation rate (β). Levels of zygotic transcripts (Z, blue) are determined by a linear accumulation, with a constant accumulation rate (α). Total mRNA levels (gray) are the sum of maternal and zygotic mRNA. Model fits (solid lines) to interpolated zygotic (blue dots) and maternal (red dots) expression levels (y-axis, log₂ scale) across 11 pseudotime bins (x-axis) for key developmental genes. Gene name and predicted accumulation pseudo-rate are indicated. Light blue and light red dots represent estimated zygotic and maternal mRNA levels, respectively, for three sampled timepoints (4.3 hpf, 4.8 hpf and 5.3 hpf). **B** Zygotic regulators. **C** Maternally provided transcripts. **D** Histogram for number of genes (x-axis, fraction) that fit only a maternal model (top), only a zygotic model (middle) or both (bottom). Numbers and percentages are indicated. **E** Distribution of two parameters of the zygotic

accumulation model across genes. Left: accumulation pseudo-rate (transcripts/pseudo-min, x-axis) per gene (y-axis, fraction). Right: transcription onset time (pseudo-min, x-axis) per gene (y-axis, fraction). Time of earliest pseudotime bin in our data (240 min.) is indicated. Distributions are shown for all genes (black), zygotic only genes (blue) and combined maternal and zygotic genes (gray). **F** Model fits (as in **B**) for genes with very low (left) or very high (right) accumulation pseudo-rates. **G** Distribution of three parameters of the maternal decay model across genes. Left: degradation pseudo-rate (half-life, pseudo-min, x-axis) per gene (y-axis, fraction). Middle: degradation onset time (pseudo-min, x-axis) per gene (y-axis, fraction). Time of earliest pseudotime bin in our data (240 min) is indicated. Right: initial expression level (log₂, x-axis) per gene (y-axis, fraction). Distributions are shown for all genes (black), maternal only genes (red) and combined maternal and zygotic genes (gray). **H** Model fits (as in B) for genes with very short (left) or very long (right) half-lives.

from single-cell metabolic labeling data and successfully predict expression changes during a dynamic spatio-temporal response.

### Regulatory rate variation shapes gene expression patterns

Our earlier analyses had revealed qualitatively that different maternal transcripts are degraded at different rates (e.g., comparing *cth1* and *aif1l* turnover, Fig. 3D). However, using the parameters inferred by our kinetic models, we can systematically study transcription and degradation differences between all genes measured in this work.

Model fitting results refined and extended the classification of genes. A subset of 2201 genes (43%, Fig. 4D) fitted only a maternal

degradation model, suggesting these are predominantly maternally contributed with minimal evidence for their zygotic transcription. Another 634 genes (12%, Fig. 4D) fitted only the zygotic accumulation model, suggesting they are zygotically transcribed, with no evidence for any significant maternal contribution. Finally, 2266 genes (45%, Fig. 4D) fitted both models, suggesting these genes are both maternally provided and zygotically expressed during our measurement window. Overall, our models estimated degradation of maternal copies for 4467 genes (88%) and accumulation of zygotic copies for 2900 genes (57%).

Estimated zygotic accumulation pseudo-rates (2900 genes, Fig. 4E) varied by more than an order of magnitude between genes

(median of 4.8*10$^{-4}$ transcripts/pseudo-min) and were highly correlated to expression levels. As expected from an accumulation model, faster production would result in accumulation to a higher level during a set time period. For example, genes such as *cdh2* and *snx12* accumulated more slowly (transcripts/pseudo-min <2*10$^{-4}$, Fig. 4F) while genes such as *cdx4* and *sox32* accumulated faster (transcripts/pseudo-min >3*10$^{-3}$, Fig. 4F). Our model also suggests that for a significant number of genes (1608, 55%, Fig. 4E) accumulation of zygotic copies started before our sampling window (onset <240 pseudo-min). We did not observe any significant differences in accumulation rates between zygotic-only and maternal-zygotic genes. However, the 149 genes with cell-type-restricted zygotic mRNA expression (Fig. 3C), had higher than average accumulation rates (median of 1.7*10$^{-3}$ transcripts/pseudo-min).

Estimated maternal pseudo-half-lives of different genes (4467 genes, Fig. 4G) ranged within two orders of magnitude (median of 32 pseudo-min). For example, genes such as *pnrc2* and *cth1* decayed quickly (half-life <15 pseudo-min, Fig. 4H) while genes such as *taf15* and *ywhaz* decayed slowly (half-life > 250 pseudo-min, Fig. 4H). On average, pseudo-half-lives of transcripts of maternal-only genes were slightly shorter than those of maternal transcripts of maternal-zygotic genes (61 pseudo-min. and 75 pseudo-min. respectively). Initial levels of maternal-only genes were also 2-fold lower on average than those of maternal-zygotic genes, while onset time of their decay was similar (263 pseudo-min on average).

These represent quantitative measurements of the degradation rate of maternal transcripts when their destruction is simultaneously obscured by zygotic replacement. Lack of notable global distinctions between regulatory rates of maternal-only or zygotic-only genes and genes that are both maternally and zygotically expressed, suggests that similar pathways act to regulate transcription and degradation in all groups. Transcription and degradation rates can be tenfold or more different between genes, highlighting their fine-tuned and precise regulation.

## Coordinated dynamics shape spatial gene expression patterns

For the 2266 genes (45%) that are both maternally provided and zygotically expressed in our experiment, mRNA levels are affected both by degradation of pre-existing copies and accumulation of new ones. Interestingly, although mRNA levels of zygotic-only genes increased over pseudotime (positive fold-change) and those of maternal-only genes decreased (negative fold-change), mRNA levels of genes with both maternal and zygotic expression had much lower fold change values (Fig. 5A). However, the distribution of their corresponding separate maternal and zygotic mRNA fold changes mirrored that of maternal-only and zygotic-only transcripts, respectively (Fig. 5A). This suggests that many transcripts are undergoing regulated replacement to maintain overall expression levels as new zygotic transcripts replace degraded maternal ones. Moreover, it emphasizes that total mRNA levels in this group obscure the dynamics of that replacement. Labeling and downstream deconvolution of maternal and zygotic transcripts are necessary to properly measure them.

To analyze in depth the temporal dynamics of maternal-zygotic genes, we categorized genes into four groups (A–D, Fig. 5B) based on the rate of destruction of their maternal copies and rate of accumulation of their zygotic copies (Methods). The 519 genes (23%) in group D combine slow destruction of maternal copies and fast accumulation of zygotic copies (e.g., *hmgn2*, Fig. 5C), resulting in overall accumulation. This combination supports high expression levels for genes in this group, which are on average 4-fold higher than in other groups (Fig. 5D). Genes in this group are enriched (Supplementary Data 2) for RNA metabolic functions, such as RNA splicing ($p < 7*10^{-35}$) and translation ($p < 2*10^{-52}$), suggesting that embryos increase their capacity to process newly transcribed mRNAs as part of the process of initiating zygotic transcription. On the other hand, the 183 genes (8%)

in group B, combine a fast destruction of maternal copies and slow accumulation of zygotic copies (e.g., *chac1*, Fig. 5C), resulting in a decrease in total mRNA levels of genes in this group over time (1.7-fold decrease, on average), and overall lower expression levels (Fig. 5D). This group is the only group enriched for components of the histone modification pathway ($p < 3*10^{-7}$).

Unlike expression levels of genes in groups B and D, which changed as a result of imbalanced destruction and replacement rates, expression levels of genes in group A and C remained relatively constant. The mean expression levels of genes in groups A and C are similar (Fig. 5D), but their underlying replacement dynamics of maternal with zygotic transcripts differed markedly. Transcripts in group C (162, 7%) exhibited slow replacement, combining slow destruction of maternal transcripts with slow accumulation of their zygotic counterparts (e.g, *faub*, Fig. 5C). Conversely, genes in group A (284, 13%) have fast replacement, combining fast degradation of maternal copies with fast accumulation of zygotic copies (e.g, *cth1*, Fig. 5C).

While the slow replacement in group C represents an energetically favorable strategy, it is not clear why genes in group A are quickly replaced. We therefore tested for differences in the expression of genes that might suggest why. Indeed, group A is the only group enriched with genes with cell-type-restricted expression (16/34 genes shown in Fig. 3C, $p < 7*10^{-7}$, hypergeometric). The combination of fast maternal degradation and fast zygotic accumulation in group A helps rapidly eliminate ubiquitous maternal expression and establish cell-type-specific expression. Additionally, Group A is enriched (Supplementary Data 2) for genes with tightly controlled temporal expression, such as cell-cycle genes ($p < 8*10^{-5}$), whose expression is restricted to specific phases of the cell-cycle (e.g., *ccnb1* at G2/M phase, *esco2* at S phase, Fig. 5E). The fast replacement kinetics of these genes possibly reflects that their mRNAs are degraded and transcribed anew each cell cycle. Since cells in our data are not analyzed with regard to cell cycle progression, such genes will seem to be constantly expressed with high degradation and transcription rates.

Overall, this analysis reveals how coordination between transcription and degradation affects gene expression. For many genes, kinetics of replacement between maternal and zygotic mRNAs result in overall increases or decreases is total mRNA over time. But even when these processes are matched and genes maintain steady expression levels, the underlying fast or slow rates have functional consequences for gene regulation. Slow rates conserve resources, but a rapid replacement helps to restrict expression of genes to a specific cell type or time.

## Lineage-specific decay rates from developmental trajectories

Transcription rates often differ between cell types, contributing to cell-type-specific gene expression patterns; a key question is whether maternal mRNAs exhibit different degradation rates between cell types that may also play such a role. We used URD to assign cells in our data to cell-type-specific developmental trajectories. Then, we applied our kinetic modeling approach to infer maternal and zygotic regulatory rates along five separate cell-type-specific developmental trajectories, for which pseudotimes assigned to cells span a sufficiently large temporal interval (Methods, Fig. 6A, Supplementary Fig. 7A, B). We used a "trajectory specific" model that assumes that regulation can differ for a specific trajectory, and thus fits two separate sets of parameters for each gene: one to fit the "trajectory-specific" profile, and another to fit all other cells not assigned to that trajectory. We compared this "trajectory specific" model to a "uniform" null model, which assumes similar rates across all cells. Genes that confidently ($p < 0.05$) reject the "uniform" model in favor of the alternative "trajectory specific" model represent cases where the dynamics of that mRNA are significantly different within one trajectory. In all other cases, we retain the "uniform" model. Given the lower numbers of cells within

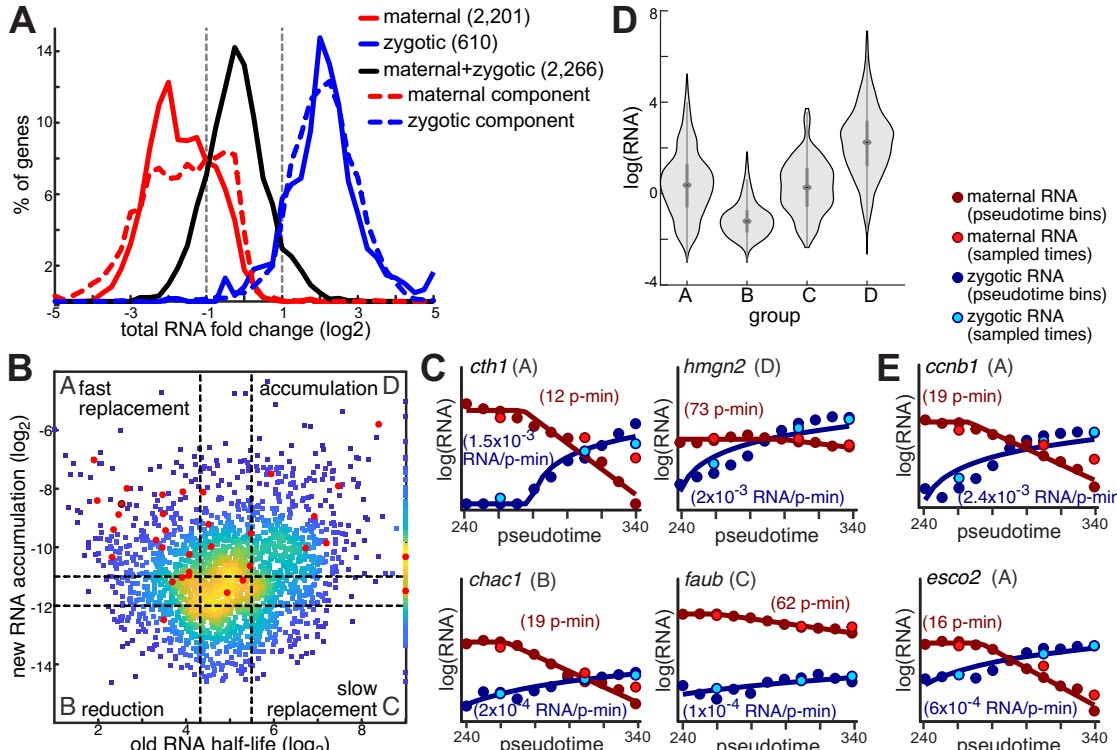

**Fig. 5 | Different replacement strategies between maternal and zygotic copies of embryonic genes underlie overall mRNA levels with fewer changes.**
**A** Distribution of total RNA fold-change (x-axis, log2) between average expression in three first pseudotime bins and three last pseudotime bins, of maternal-only (solid red), zygotic-only (solid blue) and maternal-zygotic (solid black) genes, as well as the separate maternal (dashed red) and zygotic (dashed blue) components of total maternal-zygotic RNA levels. Positive fold-change values represent an increase and negative values represent a decrease in mRNA levels over time. **B** Scatter plot of maternal mRNA half-lives (x-axis, log2) and zygotic mRNA accumulation (y-axis, log2) for maternal-zygotic genes. Colors represent density (yellow = high density; blue = low density). Red dots indicate genes that are included in the set of 154 genes with a cell-type-specific expression. Dashed lines represent lower and upper bounds for defining four groups of genes within this set A–D (half-life bounds: <20 pseudo-min or >45 pseudo-min; log2 accumulation bounds: <−12 or >−11). **C** Model

fits for one gene in each group (as indicated). Plots show model fits (solid lines) to interpolated zygotic (blue dots) and maternal (red dots) expression levels (y-axis, log2 scale) across 11 pseudotime bins (x-axis). Gene names, accumulation pseudo-rates and pseudo-half-lives are indicated. Light blue and light red dots represent estimated zygotic and maternal mRNA levels, respectively, for three sampled timepoints (4.3 hpf, 4.8 hpf and 5.3 hpf). **D** Distribution of total mRNA expression levels (y-axis, log2) for genes in each of the groups A-D (x-axis). The central dot is the median; gray box bounds are 25th and 75th percentiles, upper and lower limits of whiskers are upper and lower limits of whiskers are 1.5x interquartile ranges. Values outside of the upper and lower limits are defined as outliers, n = 284, 183, 162, and 519 genes in groups A−D, respectively. **E** Model fits (as in **C**) for two genes with fast replacement in group A, whose expression is restricted to specific phases of the cell cycle.

individual trajectories, we conservatively analyzed only a subset of genes with a significantly high expression within and outside each trajectory (Methods, Supplementary Fig. 7C).

Most genes (2196/2345, 94%) retained the "uniform" regulation hypothesis, suggesting that differences between cell lineages contribute minimally to shaping their mRNA dynamics. However, 149 genes had evidence for lineage-specific mRNA dynamics (Fig. 6B, Supplementary Data 3), indicating that lineage-specific regulation significantly contributes to shaping their expression levels, either through changes to maternal degradation or zygotic transcription.

When considering regulation of maternal mRNAs, we identify lineage-specific stabilization of 15 maternal genes. These were exclusively in the enveloping layer (e.g., *gclm, epcam*, Fig. 6C), supporting our previous observation for utilization of maternal stabilization in this cell type, and identifying additional mRNAs that are differentially degraded in this cell type. Unfortunately, as the germ-cell lineage did not span a large enough temporal interval to be included in this analysis, we could not test our previous observations of maternal stabilization in germ cells in this analysis. Lineage-specific destabilization of maternal mRNAs was not investigated by our previous cell-type enrichment analysis (Fig. 3C), and could represent an additional

regulatory layer. We find evidence for maternal destabilization of ten genes, including four in the enveloping layer (e.g., *pfn1, dhrs13a.2*, Fig. 6D) and five in the ectoderm (e.g., *hnrnpa0b*, Fig. 6D).

When considering regulation of zygotic mRNAs, we find evidence for faster lineage specific accumulation of zygotic copies of 93 genes, including 59 out of 123 genes that we previously identified with cell-type-specific expression in one of these lineages (Fig. 3C). Faster accumulation is mostly evident in the enveloping layer (68 genes, e.g., *cldnb, krt92*, Fig. 6E), and prechordal plate (22 genes, e.g., *nog1, fzd8b*, Fig. 6E). Another 43 genes showed evidence for slower accumulation of zygotic copies in specific lineages. These were mostly genes that are enriched in other lineages. Finally, four enveloping layer genes show lineage-specific regulation of both their zygotic and maternal copies, including three genes with faster zygotic accumulation and faster maternal degradation in enveloping layer (e.g., *pfn1, dhrs13a.2*, Fig. 6D). These are examples of lineage-specific fast replacement between maternal and zygotic copies, which could help establish precise expression levels within specific cell types.

To experimentally test differential stabilization of maternal mRNA within the enveloping layer, we generated and injected reporter mRNAs with UTRs that were "neutral" or from the *epcam* gene, which

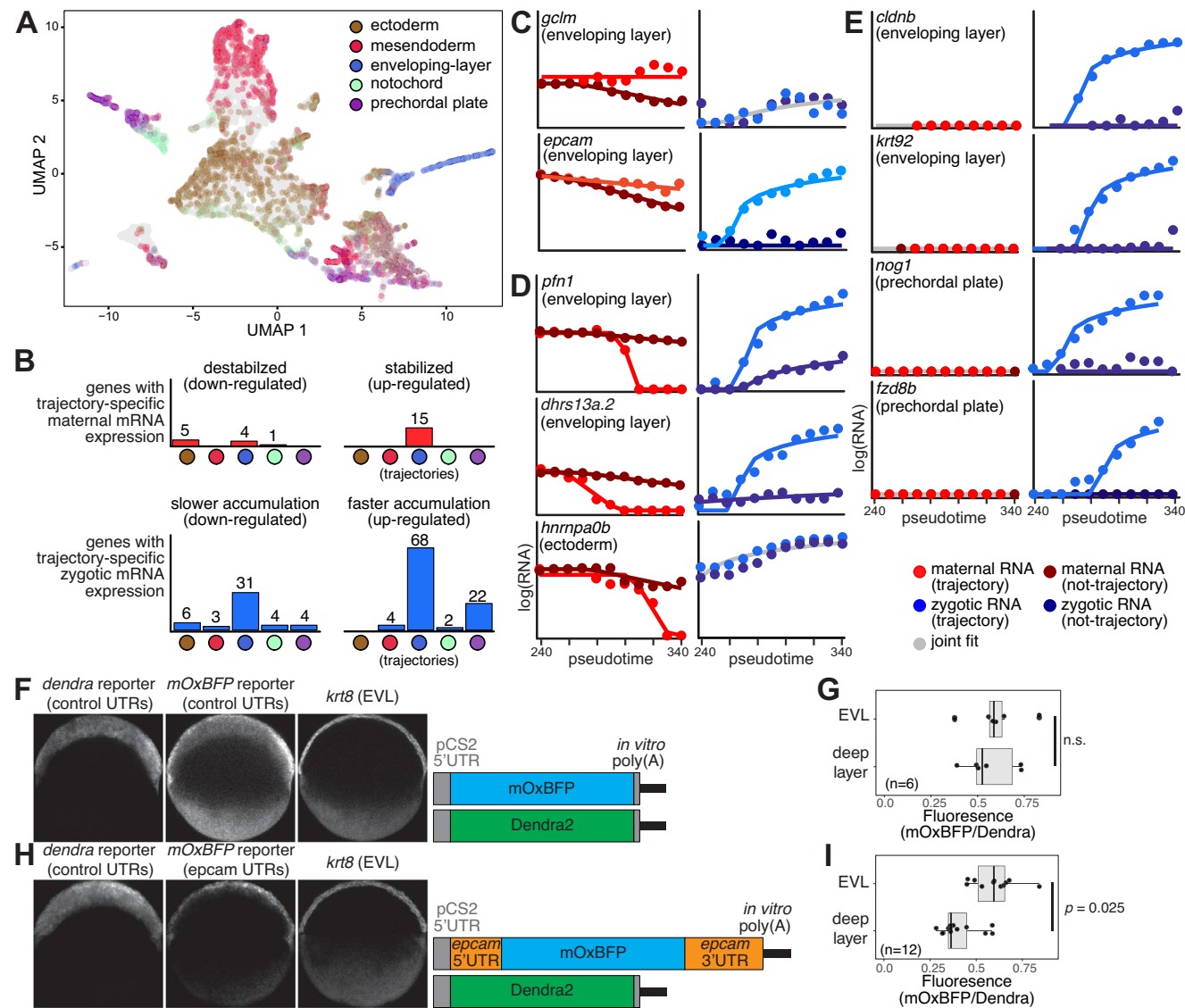

**Fig. 6 | Trajectory specific regulatory kinetic rates of embryonic genes. A** A UMAP projection of 8226 single cells with five cell lineages (colors as indicated). Each cell is colored by its lineage assignments. **B** Histograms (y-axis; number of genes) of trajectory-specific maternal (red, top) and zygotic (blue, bottom) significantly differentially regulated genes, per trajectory (x-axis, color-coded as in (**A**)). Each set of trajectory-specific genes is divided into up-regulated (right) and down-regulated (left) genes. Some genes are differentially regulated in more than one trajectory. **C**–**E** Trajectory-specific model fits (solid lines) to interpolated zygotic (right) trajectory (light blue dots) or non-trajectory (dark blue dots) and maternal (left) trajectory (light red dots) or non-trajectory (dark red dots) expression levels (y-axis, log₂ scale) across 11 pseudotime bins (x-axis) for genes with trajectory specific regulation. Expression levels were interpolated separately for cells that are assigned to a trajectory or those not assigned to it across 11 pseudotime bins. Gene name and trajectory are indicated on top. Gray lines represent fits that match both trajectory and non-trajectory data, and therefore retain the null hypothesis of similar regulation within and outside a trajectory. **F** Left: hybridization chain reaction (HCR) in situ against *dendra* mRNA with "neutral" UTRs, *mOxBFP* mRNA with "neutral UTRs", and *krt8* that marks enveloping layer (EVL) cells. Right: diagrams of injected reporter mRNAs. **G** Quantification of ratio of *mOxBFP* to *dendra* HCR fluorescence in the enveloping layer (EVL) and deep layer (rest of the blastoderm) in "neutral" UTR reporter injections from *n* = 6 independent embryos. Box plots show median (line), 1st and 3rd quartiles (hinges), and 1.5 inter-quartile ranges (whiskers). Significance was calculated using a two-sided Student's *t* test. Each sample was measured twice in the two different regions. **H**, **I** Similar to (**F**, **G**), except using *mOxBFP* mRNA with *epcam* UTRs in *n* = 12 independent embryos.

exhibited slower degradation in the enveloping layer in our kinetic models. mRNA encoding Dendra2 with "neutral" 3′ and 5′ UTRs was injected into 1-cell embryos alongside mRNA encoding mOxBFP with either "neutral" or *epcam* 3′ and 5′ UTRs. Remaining reporter mRNA was visualized at 50% epiboly (~5.3 hpf) via hybridization chain reaction alongside a marker of the enveloping layer (*krt8*). *mOxBFP* with "neutral" UTRs (Fig. 6F) was inherently slightly more stable in the enveloping layer compared to *dendra* with similar "neutral" UTRs (Fig. 6G). However, *mOxBFP* with *epcam* UTRs (Fig. 6H) was significantly more stable in the enveloping layer than the rest of the embryo (Fig. 6I). This suggests that sequence elements within the

*epcam* UTRs indeed confer differential rates of destruction within enveloping layer cells than the rest of the embryo.

Our results reveal that most lineage-specific regulation of mRNA levels happens by zygotic transcription, as expected. However, in addition to the primordial germ cells (as previously discussed), they highlight differential regulation of maternal mRNAs in the enveloping layer, by lineage-specific changes to their stability. This phenomenon has not been previously documented in enveloping layer cells, and suggests that lineage-specific alteration of maternal mRNA destruction may be shared by the earliest cell types specified in the zebrafish embryo[11].

### Linking mRNA sequence elements to differences in decay

Taking advantage of the broad and quantitative mRNA degradation rates obtained by our models, we analyzed sequences of maternal transcripts, and associated elements within them with differences in degradation. We systematically tested all 3- to 8-nt-long sequences (k-mers) within annotated UTRs for their association with differences in pseudo-half-lives and onset times estimated by our kinetic models (Methods). We restricted our analysis both by significance (Kolmogorov-Smirnov FDR < 1%) and effect size (normalized fold-difference).

Several expected[32] regulatory elements were enriched within 3′ UTRs. For example, in transcripts with short half-life, we identified the miR-430 seed (GCACUU, $p < 6*10^{-18}$, Fig. 7A), Pumilio binding sites (UGUAUAU, $p < 8*10^{-3}$, Fig. 7A) and the AU-rich consensus (UAUUUA, $p < 2*10^{-3}$, Fig. 7A). Within transcripts with delayed onset of degradation, indicative of early stabilization, we identified stabilizing poly-U signals (UUUUUUUU, $p < 2*10^{-8}$, Fig. 7B).

We also detected previously unrecognized regulatory sequences within 3′UTRs. In particular, a delayed onset of degradation is associated with several A-rich sequences such as AGAAA ($p < 1.5*10^{-4}$, Fig. 7B) and AAAAC ($p < 2*10^{-4}$, Fig. 7B), as well as other sequences such as UUCAA ($p < 1.5*10^{-4}$, Fig. 7B) and UCUGAU ($p < 2*10^{-4}$, Fig. 7B). Interestingly, A-rich sequences are also associated in our analysis with a shorter half-life (AAAAG $p < 1.3*10^{-3}$, AGAAAAAA $p < 7.6*10^{-3}$, Fig. 7A).

In addition to 3′UTR sequences, other sequence features within mRNAs can affect mRNA degradation. First, analysis of polyA tail lengths (as measured by ref. 33) showed that longer half-lives are associated with longer tails as measured at 4 hpf (Kolmogorov-Smirnov 1% FDR $p < 6*10^{-29}$, Supplementary Fig. 8A). Second, efficient translation is also associated with mRNA stability. Longer half-lives were associated with higher predicted codon optimality index[34] (Wilcoxon $p < 2*10^{-25}$, Supplementary Fig. 8B) and translation efficiency estimated by ribosomal footprinting[35] ($p < 8*10^{-18}$ at 4 hpf, Supplementary Fig. 8C). Enrichment of AUG triplet in 5′UTR sequences of genes with lower mRNA stability ($p < 1.7*10^{-14}$, Fig. 7C) could further represent the effect of reduced translation efficiency by upstream reading frames. Finally, mRNA base modifications were also linked to stability. Both m6A modifications[36] and m5C modifications[37] were associated with shorter half-life at early time-points (0–4 hpf, Supplementary Fig. 8D, E), and with longer half-life at later timepoints (6–8 hpf). Thus, an early modification is associated with faster decay, but a late modification is associated with a slower decay of the maternal transcripts.

We also analyzed 3′UTR sequences of maternal mRNAs that were stabilized within germ cells. First, we reanalyzed our complete set of 436 cell-type-specific genes, and identified 8 germ-cell-enriched maternal transcripts (Supplementary Fig. 9A, B). Only one of these genes (*surf2*) showed any evidence of zygotic RNA transcription in our data. Interestingly, maternal copies of some of these genes (e.g., *dnd1*, *nanos3*, Supplementary Fig. 9A) are undetectable in single cell profiles of somatic cells, while other genes (e.g., *gra*, *ddx4*, Supplementary Fig. 9B) also show residual expression in somatic cells, as was also previously noted[4,38]. Analysis of published RNA-Seq data[6] of sorted germ-cell populations at dome stage (4.5 hpf) further confirmed that *nanos3* and *dnd1* are highly enriched in germ-cells (160 and 100-fold, respectively), while *ddx4* and *gra*, had a lower enrichment (45 and 29-fold, respectively). Therefore, we next searched for common sequences within the 3′UTRs of genes in each of these two groups. The miR-430 seed sequence (GCACUU)[39] occurred both in germ-cell specific (3/5 genes) as well as in those with residual somatic expression (1/3 genes). However, the Dnd1 binding sequence (UUUGAUUU)[40] occurred in 3′UTRs of all germ-cell specific genes (5/5 genes) but none of the genes with residual somatic expression. This suggests that protection by Dnd1[40] is associated with maternal germ-cell markers that are completely depleted from somatic cells. However, other mechanisms to stabilize maternal transcripts in the germline, may work in a way that allows residual expression in somatic cells.

Finally, we looked for sequences that were over-represented in 3′ UTRs of stabilized transcripts within the enveloping layer. Interestingly, our analysis identified AU-rich elements (UAUUUAUU) as the most over-represented k-mers in this group (9/15 genes, Kolmogorov-Smirnov $p < 1.5*10^{-4}$; due to the small number of genes, an FDR correction was not applied).

These results confirm the association of distinct regulatory elements with differences in regulatory pseudo-rates of embryonic genes, including both expected and previously unrecognized signals. They distinguish two distinct residual expression patterns in somatic cells for maternal transcripts that are stabilized within the germline; but only one of which aligns with previously known sequence determinants of mRNA stability in those cells. These demonstrate our ability to utilize refined quantification by our kinetic modeling in order to generalize regulatory effects and identify sequence signals that encode them.

## Discussion

In this work, we globally analyze mRNA transcription and degradation dynamics during cell type specification in zebrafish embryos. We demonstrate the technological integration of metabolic labeling with scRNA-Seq in cells of a developing organism, sampled along a developmental timecourse. We develop kinetic models that integrate the analysis of timecourse data and generate a quantitative view of the relative contributions of mRNA transcription and degradation to gene regulation in the early zebrafish embryo. These models reveal the regulatory functions and cell-type-specific variations that govern gene expression programs during a spatio-temporal dynamic response. Our portal (https://liorf.shinyapps.io/zebrafish_single_cell_regulation) provides the scientific community with ready access to our data and analysis results. We highlight three main regulatory principles.

Our models uncover fine-tuned and precise differences in regulation of mRNA transcription and degradation between genes that shape their spatio-temporal expression patterns. We generate precise and quantitative measurements of mRNA transcription and degradation rates of embryonic genes both globally and within individual cell types. This has been difficult to measure when their destruction is simultaneously obscured by zygotic transcription. As expected, most cell-type-specific expression relies on the restricted zygotic transcription of genes. However, we find evidence for regulated stabilization of maternal mRNAs in two cell lineages: germ cells and enveloping layer cells. Both these lineages are specified early during zebrafish development[11], when transcription is not yet available as a tool to establish cell-type specificity. The yolk syncytial layer is another layer that forms similarly early in development and might have similar regulation of maternal RNAs. This tissue was not profiled in our study, but future single-nucleus RNA-Seq could allow study of this syncytial tissue, which is not divided into cells. Globally, regulation of maternal-only or zygotic-only genes is comparable to that of maternal-zygotic genes, suggesting that similar mechanisms regulate all different classes.

We show that coordination between transcription and degradation of maternal-zygotic genes maintains their overall mRNA levels, but that these parameters are tuned differently for each transcript. For example, slow exchange of maternal and zygotic copies of many housekeeping genes allows embryos to conserve and utilize pre-existing maternal mRNAs that are still needed later in development. On the other hand, fast exchange of maternal and zygotic copies helps to restrict zygotic expression to a specific cell type (e.g., *cth1* in muscle and blood, *cldn7b* in the enveloping layer) or time (e.g., *ccnb1* in G2/M phase of the cell cycle). Preexisting maternal copies are targeted for fast removal, and allow quick establishment of cell-type-restricted or

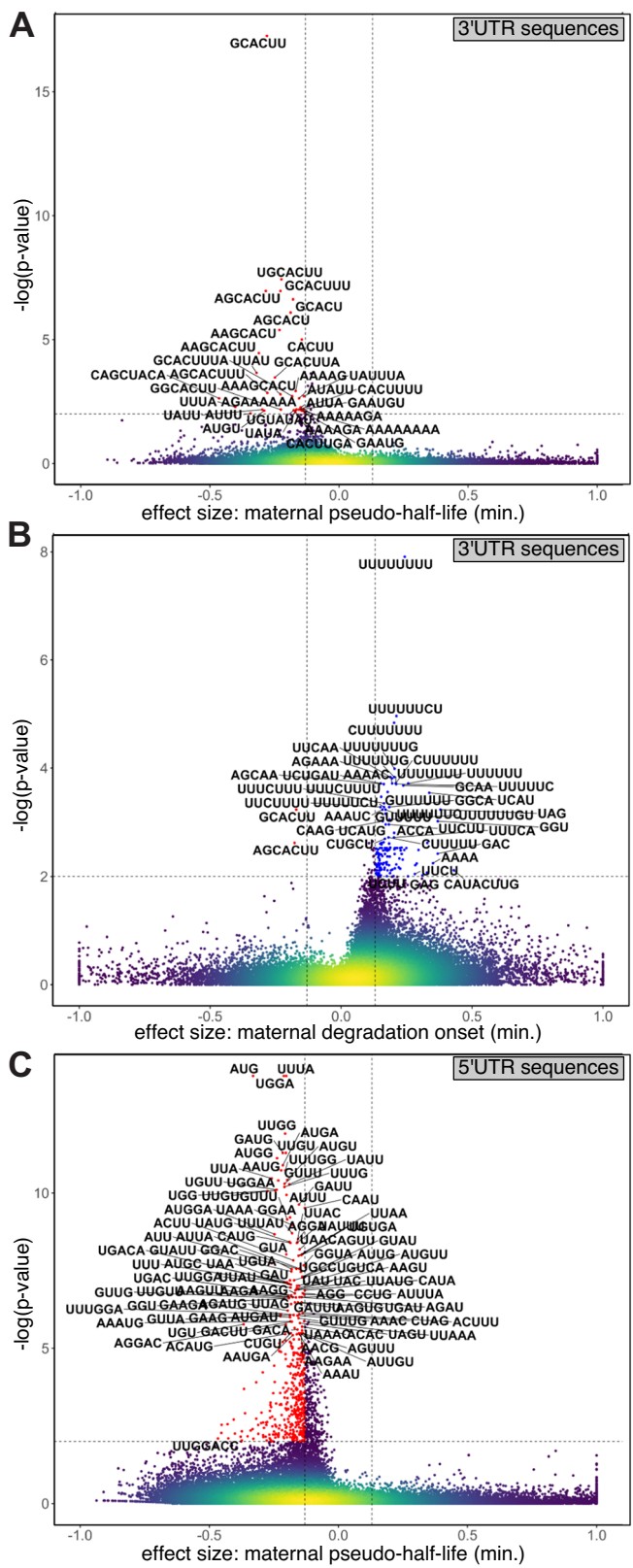

**Fig. 7 | Sequence enrichments associated with regulatory rates of mRNA degradation.** Volcano plots for the enrichments of short sequences of length 3–8 bases (k-mers) based on difference between estimated parameters for genes that include this short sequence in their sequence and those that do not. Plots show the significance (y-axis, −log$_{10}$($p$ value), one-sided Kolmogorov-Smirnov FDR < 1%) and the effect size (x-axis, difference in the standard mean of each of the two distributions). Horizontal dashed line is 1% FDR, vertical dashed line is an absolute effect size of 0.13. **A** Enrichments in longest annotated 3′UTR for the maternal pseudo-half-life parameter (log$_2$). **B** Enrichments in longest annotated 3′UTR for the maternal degradation onset time parameter. **C** Enrichments in longest annotated 5′UTR for the maternal pseudo-half-life parameter (log$_2$). Colors represent density (yellow = high density; blue = low density). Dashed lines represent thresholds for significance. Top short sequences are noted on plots.

genes with sequence similarity[42]. It is intriguing to speculate that similar mechanisms could couple between transcription and degradation of maternally inherited mRNAs.

Finally, we associate known and previously unrecognized elements within 3′UTRs of maternal mRNAs with differences in their degradation rate or timing as quantified by our models. For instance, miR-430 seeds, Pumilio binding sites, AU-rich elements, and A-rich sequences are all associated with shorter pseudo-half-lives; while poly-U signals, A-rich sequences, and some others (UUCAA, UCUGAU) are associated with delayed onset of degradation. These suggest a combined regulation of maternal mRNA stability by several pathways. We also analyze the role of 3′UTR elements relative to cell-type-specific changes in mRNA stability. We show that maternal germ cell markers with a Dnd1 binding site[40] are completely depleted from somatic cells (*e.g.*, *dnd1*, *nanos3*). However, other markers have residual non-specific expression outside the germline (e.g., *gra*, *ddx4*). In those cases, other germ-cell RNA binding proteins such as Dazl-induced cytoplasmic polyadenylation[43,44] could be involved. Interestingly, AU-rich elements were over-represented in 3′UTRs of stabilized transcripts within the enveloping layer. Typically, AU-rich elements destabilize mRNAs, raising the possibility that this process is somehow attenuated within the enveloping layer. Several RNA binding proteins exhibit enveloping layer-specific expression during blastula stages in our dataset; among them, *ptbp1a* has been linked to mRNA stabilization by binding to ARE sequence elements in 3′UTR[45,46], suggesting it may play a role in the enveloping layer.

The technical framework of this work is reproducible and reliable by several tests but could be further improved and expanded in certain directions. In particular, concentrations of 4sU that are tolerated by cells only replace a relatively low number of uridines by 4sU (at most one in ten uridines in our samples, Supplementary Fig. 2A). Thus, statistical inference has more limited accuracy within single cells with fewer reads, and can underestimate labeled mRNA fractions in some cases. This problem is partially mitigated by performing pseudo-bulk analysis of scRNA-seq data. Improved statistical modeling that aggregates similar cells within a sample could further reduce such biases. Interpolation of temporal information by pseudotime is also limiting, and allows inference of pseudo-rates rather than absolute values. Pseudotime units, which are estimated from transcriptomic changes, might also correspond to non-uniform time units. However, as all genes will be similarly influenced by such biases, differences between genes will not be significantly affected. Optimizing the joint likelihood of rate parameters across all genes could further improve estimation accuracy. A linear correlation between pseudo-rates of maternal-only genes and absolute rates measured in bulk RNA-Seq (Supplementary Fig. 10), suggests an approach to correct pseudo-estimates and indirectly infer absolute time. Finally, modeling within separate developmental trajectories is limited by temporal intervals. Measuring later developmental stages, when cell lineages have further differentiated, will allow more accurate quantification of differences in mRNA

temporally restricted zygotic expression and eliminate background. A similar mode of regulation was also observed during fly development[41]. It is unclear why a few cell-type-specific genes implement slow exchange of maternal and zygotic copies, which obscures their cell-type-specific expression until later in development (e.g., *aif1l* in muscle and blood). It was recently shown that mRNA destabilization in embryos can trigger compensatory changes in the transcription of

regulation. In addition, tools to control the timing of metabolic labeling within embryos will uncover the effect of mRNA stability of zygotic genes that drive later developmental events.

Our work systematically dissects quantitative mRNA transcription and degradation rates in vivo within developing embryos with an unprecedented cell type and temporal resolution, and learns their regulatory principles that define and maintain gene expression programs during a dynamic spatio-temporal response.

## Methods

### Zebrafish
All protocols and procedures involving zebrafish were approved by the Harvard University/Faculty of Arts and Sciences Standing Committee on the Use of Animals in Research and Teaching (IACUC; Protocol #25-08), the Hebrew University Ethics Committee (IACUC; Protocol #NS-15859), and the National Institutes of Health (ACUC, Protocols 20-001 and 23-001). Embryos were grown and staged according to standard procedures[27]. Zebrafish embryos from wild-type AB/TL strains were used for all experiments. Sex was not considered in study design, as it is not determined at the early developmental stages studied in this work.

### mRNA spike-ins cloning and transcription
Constructs encoding either a GFP or an RFP protein were PCR amplified using the following primers: F 5′–gaaatacgactcactatagggggccacccaagctcatcgattcgaattcatg–3′ and R 5′– gcggccgcagacatgataag–3′, and in vitro transcribed from a T7 promoter with the HiScribe T7 Kit (NEB), using a mixture of ATP, CTP, GTP and UTP in transcription of GFP, or replacing UTP with 4sUTP (TriLink Biotechnologies) in transcription of RFP, to produce RFP mRNAs that were in vitro transcribed with 100% 4sU residues. Resulting mRNA levels were quantified by Qubit fluorometric quantification (Thermo Fisher) and mRNA length was validated by gel electrophoresis. For barcode mixing controls, mRNAs encoding mCherry with C-terminal fusions of either SV40 or nucleoplasmin nuclear localization signals were transcribed from pCS2-SV40nls-mCherry-SV40nls or pCS2-SV40nls-mCherry-NPLnls plasmids, using the mMessage Machine SP6 kit (Ambion/ThermoFisher).

### Fish microinjection and single-cell sample collection
Fertilized eggs were collected at 28 °C, and kept in culture medium (5.03 mM NaCl, 0.17 mM KCl, 0.33 mM CaCl$_2$, 0.33 mM MgSO$_4$, 0.1% Methylene blue). One-cell staged wild-type zebrafish embryos were removed from their chorion and injected with 1 nL of a solution containing 20 mM 4sUTP (TriLink Biotechnologies), 30 ng/µL GFP mRNA, 30 ng/µL fully-labeled RFP mRNA and 5 ng/µL of either mCherry-SV40nls or mCherry-NPLnls mRNA. For the first replicate, a total of 100 injected embryos were randomly collected at the Dome stage after visually ensuring that all embryos were at the same expected developmental stage. Embryos were placed in 2 mL ice-cold deyolking buffer (55 mM NaCl, 1.8 mM KCl, 1.25 mM NaHCO$_3$), shaken in a thermal shaker (1100 rpm) for 30 s at 4 °C and spun down (500 g) for 1 min at 4 °C. Cells were washed three times with 1 mL ice-cold deyolking wash buffer (10 mM Tris pH 8, 110 mM NaCl, 3.5 mM KCl, 2.7 mM CaCl$_2$)), shaken in a thermal shaker (1100 rpm) for 30 s at 4 °C and spun down (500 rpm) for 1 min at 4 °C. Final pellet was resuspended in 500uL PBS. Cells were then passed through a 70 µm cell sieve. Additionally, a total of 15–25 injected embryos were randomly collected per sample at the 30% and the 50% epiboly stages, and manually deyolked and dissociated. Single embryos were visually confirmed to be at the correct stage and were transferred to Petri dishes that had been coated with 2% agarose, filled with DMEM/F12 media (Gibco/Life Technologies), and allowed to soak for 3 h. Two pairs of watchmaker forceps were used to dissect the blastula cap of the embryo away from the yolk.

First, one pair of forceps was used to hold the cap, while the other was used to cut and pinch away yolk that extended below the blastula cap. Then, the blastula cap was cut slightly up the side, and the yolk inside was gently peeled away. Several blastula caps were transferred by pipette into a microfuge tube that contained 200 µl of DMEM/F12 media. The cells were dissociated by vigorously flicking the tube ten times, and then pipetting the entire volume twice while visually confirming that dissociation had occurred. For the second replicate, a total of 70–75 injected embryos were randomly collected per sample at the 30% and the 50% epiboly stages, after visually ensuring that all embryos were at the same expected developmental stage. Embryos were then deyolked and dissociated similarly to dome-stage embryos in the first replicate.

### Fish microinjection and bulk sample collection
AB/TL embryos were injected at the 1-cell stage with 1 nl of a solution containing 0.025% phenol red, 0.1 M KCl and 50 mM 4sU (Sigma). Embryos were grown in the dark and collected at the 4-cell and 30% epiboly stages (25 embryos each). Upon collection, embryos were disrupted in TRI Reagent (Sigma). RNA was isolated according to the manufacturer's protocol, followed by ethanol precipitation. RNA was then treated with a TURBO DNA-free kit (Invitrogen) according to the manufacturer's protocol, followed by ethanol precipitation. For RNA alkylation, RNA (2.9 µg) was first preincubated with 1 mM dithiothreitol (DTT) for 10 min at 55 °C and transferred to ice. Alkylation was conducted in sodium phosphate buffer (46.6 mM Na$_2$HPO$_4$, 3.4 mM NaH$_2$PO$_4$, pH 8), 50 % dimethyl sulfoxide (Sigma) and 10 mM iodoacetamide (Sigma; freshly prepared 100 mM stock dissolved in ethanol). The reaction was incubated 15 min at 50 °C and quenched by adding DTT to 20 mM. RNA was cleaned up on an RNA Clean & Concentrator-25 column (Zymo) according to the manufacturer's protocol. For poly(A) + RNA sequencing, libraries were prepared using a KAPA Stranded mRNA-Seq Kit (Kapa Biosystems) according to the manufacturer's protocol and sequenced on an Illumina NextSeq 500 platform with 75 nt single-end reads. Sequencing of ribo-depleted RNA was performed by Macrogen. Libraries were prepared with an Illumina TruSeq Stranded Total RNA with Ribo-Zero Human/Mouse/Rat kit, and sequenced on an Illumina NovaSeq 6000 platform with 100 nt paired-end reads. Data was deposited in GEO under accession GSE224113.

### Collection of Drop-seq transcriptomes
Drop-seq droplet encapsulation was performed essentially as described in the Drop-seq protocol version 3.1 (12/28/2015, available http://www.dropseq.org/). To prevent sticking, cell suspension syringes and tubing were pre-blocked with PBS + 0.1% BSA for 1 h prior to collection. To reduce sedimentation, cells were resuspended in PBS + 0.1% BSA + 20% Optiprep, and an additional mixer magnet was added to the cell suspension, which was gently agitated by hand every 30–60 s using a bar magnet. Encapsulation was performed with standard Drop-seq microfluidic devices flushed with Aquapel (Flowjem) and standard Drop-seq beads (Chemgenes, MACOSKO-2011-10). Droplets were held on ice for up to 1 h before starting the RNA recovery process.

We added an RNA alkylation step[30] after mRNA capture on beads. Such chemically induced alkylation of 4sU residues alters base pairing during reverse transcription and creates characteristic T-to-C conversions in downstream sequencing reads, allowing quantification of labeled RNAs. Droplets were held on ice for up to 2 h (to enable multiple sequential collections) before starting the RNA recovery, chemical treatment and reverse transcription. Iodoacetamide (IAA, Sigma-Aldrich) was freshly dissolved in ethanol to make a 100 mM stock solution. Following breakage of droplets, beads were washed once with 300 µL of 5× IAA buffer (250 mM NaPO$_4$ pH8), and incubated in IAA solution (50 mM NaPO$_4$ pH8, 10 mM IAA, 20% DMSO, 6% Ficoll PM-400) for 15 min at 32 °C with rotation. DTT was added to a final concentration of 20 mM to stop the reaction, and beads were washed

twice with 6x SSC solution before continuing with reverse transcription according to standard protocol.

Libraries were built according to the Drop-seq protocol version 3.1 (12/28/2015, available http://www.dropseq.org/) with the following modifications. An estimated 50–100 STAMPs were included in each 50 μL PCR reaction, and 12–13 cycles of PCR amplification were performed. Libraries were sequenced using Illumina Nextseq v2.2 Mid or High Output 150 bp chemistry. Data was deposited in GEO under accession GSE224918.

### Alignment and quality control of Drop-seq data

**Processing of sequencing data.** Alignment of sequencing reads and generation of digital expression matrices was performed using Drop-seq tools v1.12. Ensembl Zv10 Release 82 was used as the reference genome with the spike-in transcripts used (GFP/RFP/mCherry with different C-terminal nuclear localization signals) added as an additional chromosome. Gene and transcript annotations from the Ensembl Zv10 Release 82 reference were used, but transcripts were assigned gene names by giving priority to names from ZFIN (Zebrafish Information Network) and using Ensembl names as backup. More specifically: if ZFIN and Ensembl agreed (81% of gene names fell into this category) that gene name was used; if only ZFIN or Ensembl assigned a name to the gene, that name was used; if neither ZFIN nor Ensembl had a name for the gene, the Ensembl gene ID was used. If both ZFIN and Ensembl named a gene, but the names did not agree, then the name with fewer punctuation marks was used (this, for instance, prioritizes names like RNH1 over cDNA clone identifiers like SI:CH211-66C13.1 when the two sources disagree). When Ensembl annotated several unresolved copies of a gene, e.g., RNH1 (18 of 55), the multiple copy identifier was stripped. All reads for a given gene name were combined. Since the T-to-C conversions induced by alkylation of 4sU could affect mapping rates, we made modifications to ensure that T-to-C conversions did not reduce mapping rates by converting all bases in the genome and sequencing reads. Two modified versions of the genome were produced: one where all T residues were replaced with C and one where all A residues were replaced with G. Prior to alignment, original sequencing reads were preserved and then all A residues were replaced with G. These converted reads were aligned to both modified genomes using Bowtie2 with the following parameters: --phred33 --reorder, then filtered based on which strand the read was aligned to. These two outputs were combined and then processed through the remainder of the Drop-seq tools pipeline to produce processed BAM files and digital gene expression matrices. Converted reads were then replaced with the original reads and CIGAR strings were recalculated using Samtools in the final output BAM files. While a significant number of reads was mapped to three non-labeled controls (GFP and mCherry spike-ins sequences), we detected only very few reads that mapped to the 100% 4sU-labeled RFP injected control, which did not allow to infer conversion rates for this control. We indeed validated that the IAA treatment of this 100% labeled mRNA led to a significant degradation of this mRNA.

**Identification of highly variable genes and batch correction.** First, the digital gene expression matrix output by Drop-seq tools was imported into URD[11], filtered to retain only cells with expression of at least 500 genes, and the data was log-normalized. This resulted in 8226 single cells, with 1,386 genes per cell on average (with 1 or more UMIs). 924 highly variable genes were identified using findVariableGenes (*diffCV.cutoff* = 0.2). There was a noticeable batch separation between samples that received IAA treatment and control samples, as well as between samples processed using the two different dissociation methods described above. To correct for this, gene expression modules were calculated using non-negative matrix factorization as described (Farrell et al. 2018 Science) with parameters (*k* = 35, *rand_state* = *None*, *alpha* = 2, *l1* = 0.5, max_iter = 10,000, *rep* = 5, *init* =

*random*). Six modules were identified that differed primarily between dissociation methods or treatment conditions (NMF 2, NMF 7, NMF 11, NMF 14, NMF 17, and NMF 22). Genes highly loaded into those six batch-associated modules (as determined by an elbow plot of gene loadings), as well as mitochondrial genes and spike-ins were removed from the highly variable genes, leaving 854 remaining highly variable genes.

**Determination of cell clusters.** Principal component analysis was computed using the highly variable genes (*calcPCA*). Cell type clusters were calculated with Jaccard/Louvain graph clustering (*graphClustering, dim.use* = "pca", *which.dims* = c(1:7, 9:10, 12:16), *method* = "Louvain", *do.jaccard* = T). Principal components to include in the clustering were determined based on an elbow plot and by inspection of top loaded genes to identify PCs that represented known developmental processes in zebrafish blastula. PC8 was excluded because it was primarily associated with technical quality of libraries, and PC11 was excluded because it was primarily associated with cell cycle stage. Clusters were annotated based on expression of known marker genes (*nanos3, dnd1, krt8, gsc, noto, tbxta, eve1, chrd, sox3, isg15, sesn3, osr1*) and developmental stage. Endodermal progenitors were manually separated into their own cluster based on their expression of *sox32*.

**UMAP projection.** First, a diffusion map was computed using *destiny*, as incorporated into URD (*calcDM, knn* = 50, *sigma.use* = "local"). Then, the first 25 diffusion components were used as the basis to compute a UMAP projection using the *umap* function from the R package *umap*.

**Pseudotime calculation.** Pseudotime was calculated using URD (*floodPseudotime, root.cells* = (see below), *n* = 100, *minimum.cells.flooded* = 2). URD requires user specification of the youngest cells or "the root." 100 cells were chosen to use as the 'root' (i.e. the starting point for the pseudotime calculation) based on 3 criteria: (1) they were from the earliest stage in the data (dome), (2) they were in the dome stage cluster that did not exhibit signs of specification (cluster 10, "Unspecified Dome Stage"), and (3) they had the highest expression of an NMF module (NMF33) defined primarily by exclusively maternally loaded genes (e.g., *cldng, cldnd, ccna1*, and *cth1*).

**Identification of developmental trajectories.** For a subset of cell types, developmental trajectories were calculated using URD. URD requires that the terminal cells or "tips" are defined by the user. Tips were identified from each terminal cell cluster by identifying the 12.5% of the 50% epiboly cells (minimum 25 cells) that were oldest in pseudotime. The transition matrix was biased (*pseudotimeDetermineLogistic, optimal.cells.forward* = 25, *max.cells.back* = 50) and random walks were simulated from each tip (*simulateRandomWalksFromTips, n.per.tip* = 10,000, *root.visits* = 1, *max.steps* = 1000). For each cell, its maximum number of visitations by walks from a single tip were determined. Then, cells that were visited at least 50 times (i.e. by at least 0.5% of random walks) were associated with a trajectory if they were visited during walks from that cell type tip at least 50% as often as they were visited by walks from whichever cell type visited them the most strongly.

### Analysis of nucleotide conversion signals

GRAND-SLAM (GRAND3_3.0.0)[21] was used to calculate the fraction of labeled RNA (NTR) of each gene. For analysis of bulk RNA-Seq datasets, GRAND-SLAM was used with default parameters and a gedi (GRAND-SLAM_2.0.5 f) genome reference was created using GRCz11 genome (GenBank:GCA_000002035.4) and Ensembl annotations release 102[47]. For single-cell analysis of scRNA-Seq datasets, GRAND-SLAM was used with default parameters, and gedi genome reference was created by manual additions of RFP, GFP and mCherry spike-in sequences to the GRCz11 genome. For pseudo-bulk analysis of scRNA-Seq datasets,

GRAND-SLAM was used with the following non-default parameters: -pseudobulkName clusters, -pseudobulkPurity 0, -pseudobulkFile <pb_file>, using a different file for each of the different pseudo-bulk runs (cell types, developmental stages or pseudo-time bins).

## Analysis of maternal and zygotic expression levels in single cells and UMAP visualization

Total normalized RNA expression per cell was calculated from the number of unique molecular identifiers (UMIs) using the *NormalizeData* function from Seurat version 4.1.0[48] with default parameters. Briefly, normalization divides the total number of UMIs of each gene in the cell by the total UMI counts in the cell, multiplied by 10,000 and natural-log transformed using *log1p*. For a gene $g$ at cell $c$ this is given by:

$$RNA_{total}(g,c) = \log \frac{UMI(g,c)}{\sum_{g' \in genes} UMI(g',c)} \cdot 10,000 \qquad (1)$$

Fraction of labeled RNA (NTR) estimated by GRAND-SLAM per gene $g$ in cell $c$ was used to calculate maternal and zygotic RNA expression by:

$$RNA_{maternal}(g,c) = \log \frac{UMI(g,c) \cdot (1 - NTR(g,c))}{\sum_{g' \in genes} UMI(g',c)} \cdot 10,000 \qquad (2)$$

$$RNA_{zygotic}(g,c) = \log \frac{UMI(g,c) \cdot NTR(g,c)}{\sum_{g' \in genes} UMI(g',c)} \cdot 10,000 \qquad (3)$$

We only analyzed genes which had 3 or more UMIs per cell, and a total of 50 or more UMIs across all cells, resulting in analysis of 195 genes per cell on average.

For plotting maternal and zygotic RNA expression on a UMAP projection, we first calculated a distance matrix based on the pairwise similarities between the cells' embeddings. The similarities were calculated using a gaussian kernel function with a beta parameter, which adjusts the spread of the kernel. Each of the total-RNA, maternal-RNA or zygotic-RNA normalized single cell expression matrices were multiplied by the distance matrix and divided by the sum of expression in each row. A single UMAP color-scale was defined for each gene, and used for both maternal-RNA and zygotic-RNA plots. Color-scale maximum was defined by the maximal total-RNA density value. Color-scale minimum was defined by the 30th percentile of maternal-RNA and zygotic-RNA density values across all cells with a UMIs count of 3 or more for the gene.

## Analysis of maternal and zygotic expression levels in pseudo-bulk samples

Expression values (RNA) calculated by GRAND-SLAM were normalized as follows, for gene $g$ at pseudo-sample $s$:

$$RNA(g,s) = \frac{RNA(g,s)}{\sum_{g' \in Genes} RNA(g',s)} \cdot 10,000 \qquad (4)$$

Fraction of labeled RNA (NTR) estimated by GRAND-SLAM per gene $g$ in pseudo-bulk sample $s$ was used to calculate maternal and zygotic RNA expression by:

$$RNA_{maternal}(g,s) = RNA_{total}(g,s) \cdot (1 - NTR(g,s)) \qquad (5)$$

$$RNA_{zygotic}(g,s) = RNA_{total}(g,s) \cdot NTR(g,s) \qquad (6)$$

Final expression levels were log-transformed and floored to −4.

## Analysis of total expression levels in scRNA-Seq samples

For pairwise comparisons of single-cell samples, in each of the two samples compared, we calculated a pseudo-bulk expression level for each gene $g$ by:

$$RNA(g) = \log_2 \frac{\sum_{c \in cells} UMI(g,c)}{\sum_{g' \in genes} \sum_{c \in cells} UMI(g',c)} \cdot 1,000,000 \qquad (7)$$

## Selecting genes with cell-type-restricted maternal or zygotic mRNA expression

For each gene in our data, we tested the enrichment of either its zygotic (newly transcribed) or maternal (pre-existing) transcripts, by comparing the distribution of its maternal or zygotic UMI counts within cells assigned to a specific cell type and cells not assigned to that type with a one-sided Kolmogorov-Smirnoff test, requiring higher counts within a cell type. The maternal and zygotic UMI counts per cell were calculated by multiplying the total single-cell UMI count by the fraction of labeled RNA (NTR) as calculated by GRAND-SLAM for single cells. We restricted our analysis by pseudotime, and tested the enrichment of cells within a specific cell type compared to other cells in our data of a similar pseudotime (difference of less than 20 pseudotime units). P-values were corrected for multiple hypothesis testing by a 1% Bonferroni correction. For each gene, we considered only the most significantly enriched cell type for further analysis (after Bonferroni correction).

## Dividing cells into pseudotime bins

URD-assigned pseudotime values (per-cell) were converted from arbitrary units (au) to pseudo-minutes post fertilization (pseudo-mpf) using a simple linear transformation, so that the first sample is assigned the value 240 pseudo-mpf and the last sample 360 pseudo-mpf. Cells with a pseudotime (au) smaller than 0.1 were treated as 0.1. Cells were divided into 10 pseudo-minute bins with 3 pseudo-minute overlap between adjacent bins. The last bin, which was significantly smaller than the rest, was joined with the preceding bin, creating 11 bins in total.

## Kinetic models of maternal and zygotic mRNA expression dynamics

We modeled maternal mRNA ($M$) and zygotic mRNA ($Z$) expression levels by two independent kinetic models.

### Maternal mRNA

We compare three alternative models for temporal maternal mRNA kinetics.

The simplest (null) model assumes no maternal expression of a gene, and has no parameters:

$$F_{NULL} : \log_2 M(t) = 0 \qquad (8)$$

The simpler alternative model assumes very low degradation of maternal mRNA, and has only one parameter: initial expression level ($x_0$),

$$F_1 : \log_2 M(t) = \log_2 x_0 \qquad (9)$$

The second alternative model, describes dynamic changes in maternal mRNA levels using an exponential decay with 3 parameters: a constant decay rate ($\beta$), initial expression level ($x_0$), and degradation onset time ($d$):

$$F_2 : \frac{dM}{dt} = -\beta M, \; M(0) = M(d) = x_0 \qquad (10)$$

Solving this function analytically we get:

$$M(t) = x_0 \cdot e^{-\beta(t-d)} \tag{11}$$

And we solve it looking for the log expression rate:

$$\log_2 M(t) = \log_2 x_0 - \beta(t-d) \cdot \log_2 e \tag{12}$$

## Zygotic mRNA

We compare three alternative models for temporal zygotic mRNA kinetics.

The simplest (null) model assumes no zygotic expression of a gene, and has no parameters:

$$F_{NULL} : \log_2 Z(t) = 0 \tag{13}$$

The simpler alternative model is a linear accumulation model with 2 parameters: constant transcription ($\alpha$), and transcription onset time ($d$). This model assumes very low degradation of zygotic RNA during our sampling time.

$$F_1 : \frac{dZ}{dt} = \alpha, Z(0) = Z(d) = 0 \tag{14}$$

Solving this function analytically, we get the linear accumulation model:

$$Z(t) = \alpha(t-d) \tag{15}$$

And in log-scale:

$$\log_2 Z(t) = \log_2 \alpha + \log_2(t-d) \tag{16}$$

The second alternative model adds a third parameter, using a constant degradation rate ($\beta$) to describe decay of zygotic mRNAs:

$$F_2 : \frac{dZ}{dt} = \alpha - \beta Z, Z(0) = Z(d) = 0 \tag{17}$$

Solving this function analytically, we get:

$$X_N(t) = \frac{\alpha}{\beta}\left(1 - e^{-\beta(t-d)}\right) \tag{18}$$

And in log-scale:

$$\log_2 X_N(t) = \log_2 \alpha - \log_2 \beta + \log_2\left(1 - e^{-\beta(t-d)}\right) \tag{19}$$

We fit each model separately to time-course data of either maternal or zygotic mRNA expression per gene. Fitting to $F_{NULL}$ models was defined by M(t) = − 4 and Z(t) = − 4, and any gene that had more than 4 pseudo-bins with expression lower than -3 was also fitted to this model. Fitting to non-linear models was done using non-linear least squares regression with multiple start values with R's *nls.multstart* package (version 1.0.0.), using 500 different start combinations (*iter* = 500), and parameter bounds as listed (Supplementary Table 1).

Finally, we compared the fits to each of the three nested models ($F_{NULL}$ is nested in $F_1$ which is nested in $F_2$) by a likelihood ratio test for each gene, and identified genes that fit $F_1$ and $F_2$.

The $\sigma$ for the likelihood ratio test was defined by:

$$\widehat{\sigma_{ML}} = \sigma\sqrt{\frac{n-p}{n}} = \sqrt{\frac{1}{n-p}\sum_{i=1}^{n}(y_i - \widehat{y_i})^2} \cdot \sqrt{\frac{n-p}{n}} = \sqrt{\frac{1}{n}\sum_{i=1}^{n}(y_i - \widehat{y_i})^2} \tag{20}$$

## Analysis of model accuracy by goodness-of-fit test

We estimated the fit of each gene to the kinetic model using a goodness-of-fit test, by calculating:

$$\hat{S} = \sum_{t \in T}\left(\frac{m_{g,t} - \widehat{m}_{g,t}}{\sigma_g}\right)^2 \tag{21}$$

where $M_g = \left\{m_{g,t}|t \in T\right\}$ are the observed temporal samples, $\hat{M}_g = \left\{\hat{m}_{g,t}|t \in T\right\}$ are the predictions by the model, and $pseudotime(t) = \begin{cases} 260 & t = \text{dome} \\ 310 & t = \text{epb30} \\ 340 & t = \text{epb50} \end{cases}$.

We calculated a standard deviation ($\sigma_g$) for each group of genes with a similar mean expression ($G$) by comparing their expression in two replicate samples:

$$\sigma(G) = \sqrt{\frac{\sum_{g \in G}\sum_{c \in C}(RNA(g,c,A) - RNA(g,c,B))^2}{N}} \tag{22}$$

where $C = \left\{\text{epb30,epb50}\right\}$ and $RNA(g,c,A)$ is the RNA for gene g, at stage c, in each of two replicates (A or B). Genes were divided based on the quantiles of the mean expression of maternal or zygotic RNA into 6 or 10 groups, respectively. Each group of zygotic RNA contained ~483 genes, and each group of maternal RNA contained ~446 genes.

We performed a chi square test $\hat{S} \sim \chi^2(\text{dof})$, where degrees of freedom (dof) is the number of pseudotime bins with a reliable estimated expression for the gene.

## Analysis of model fitting by simulation studies

Simulated gene expression data was generated by selecting five values for each parameter in the model, resulting in 125 sets of parameters, and applying the model to generate expression data for each set, which we term "simulated genes". For each simulated gene, we added random normal noise with mean $\theta = 0$ and variance $\sigma = sd - \frac{sd}{2}, \sigma = sd, \sigma = sd + \frac{sd}{2}$, where sd is as described above by mean expression, and repeated the procedure 100 times for each simulated gene. The data was then fitted to the kinetic models as described above. We used a goodness-of-fit test as described above to compare the fit to the original simulated data points (without noise).

## Models of maternal and zygotic mRNA expression dynamics in developmental trajectories

For model fitting, we modified the original trajectories that were calculated during single-cell analysis. First, we combined trajectories with a more than 50% overlap in cells into a single trajectory. This way the endoderm A, B and lateral plate mesoderm were combined to a single trajectory, and the neural and nonneural ectoderm to another trajectory. Second, we excluded from the analysis trajectories that did not have estimated bulk expression for at least six pseudotime bins. This resulted in a final set of five trajectories for further analysis: enveloping layer, notochord, prechordal plate, ectoderm and mesendoderm. For each of the five trajectories, we calculated pseudo-bulk dynamic expression separately for cells within the trajectory and cells not assigned to this trajectory. For each of the two groups of cells,

we divided the cells into pseudotime bins as described above, and applied pseudo-bulk calculation for each bin to estimate expression.

To this data, we applied two separate model fitting approaches. First, a "cell-type-specific" model that assumes regulation is different within the trajectory, and thus fits two separate sets of kinetic parameters for each gene: one to fit the "trajectory-specific" profile, and another to fit the "non-trajectory" profile of all other cells not assigned to that trajectory. Second, a "uniform" null model, which assumes a similar rate across all cells and thus fits a single kinetic model for both the "trajectory-specific" and the "non-trajectory" profiles. We used a likelihood ratio test to compare the two models based on the different number of parameters used for either model, and standard variation ($\sigma$) that was calculated as described above. We calculated an effect-size for the difference between the trajectory and non-trajectory maternal-RNA and zygotic-RNA expression of each gene, by the average fold-change between the trajectory and not-trajectory expression at the three pseudo-bulk samples with maximal difference. Genes were defined as "trajectory-specific", if the corrected p-value was smaller than 0.01, the fit to the trajectory specific model had an R-squared value above 0.8, and the effect-size was larger than 2 or smaller than 0.5.

## Grouping genes by combinations of maternal and zygotic kinetic parameters

For genes that are both maternally provided and zygotically expressed during the maternal-to-zygotic transition, we defined four groups of genes with distinct combinations of maternal and zygotic kinetic parameters by setting a lower-bound and upper-bound on each of the two parameters: degradation of maternal RNA (half-life bounds: <20 pseudo-min or >45 pseudo-min) and accumulation of zygotic RNA (log2 accumulation bounds: <−12 or >−11).

## Gene set enrichment analysis

Functional enrichment analysis was performed using *g:Profiler* R client (version e106_eg53_p16_65fcd97) with a 5% g:SCS multiple testing correction method[49].

Enrichment of polyA lengths[33] was analyzed by performing a one-sided Kolmogorov-Smirnov test (*ks.test*) between the polyA lengths of genes within a gene-set and all other genes in the dataset, using a 1% false discovery rate correction (FDR).

## Sequence k-mer enrichment analysis

UTR sequences were downloaded from ensembl biomart, version 103[50]. For each gene, annotated UTRs were filtered to keep a single longest annotated UTR sequence. Genes with an annotated UTR sequence below 10 nucleotides were removed from the analysis. Sequences were represented by a set of all short sequences between 3−8 nucleotides long (k-mers) using *ape* and *k-mer* R packages. We associated a k-mer with a regulatory effect when genes with this k-mer in their UTR had a significantly different distribution (one-sided Kolmogorov-Smirnov test, 1% FDR) of a specific parameter than genes without this sequence. We assigned an effect size to each k-mer by calculating the standardized mean difference defined as $\theta = \frac{\mu_1 - \mu_2}{\sigma}$, where $\mu_1$ is the mean of the first population, $\mu_2$ is the mean of the second population and $\sigma$ is the standard deviation (based on both populations).

## Analysis of additional sequence features within mRNAs that affect mRNA degradation

To calculate codon optimality, zebrafish CDS sequences were downloaded from Ensembl Biomart and filtered to include only sequences starting with a start codon, ending with a stop codon, and sequence lengths being multiples of three. Codon optimality was calculated using icodon R package[34], keeping only the highest value per gene to keep only a single value per gene. Transcript translation efficiencies at

different zebrafish developmental stages were taken from ref. 35. A gene was considered m6A modified if evidence for m6A modification was measured by ref. 36 in either m6A-seq or m6A-CLIP-seq data. A gene was considered m5C modified if it contained a location with an m5C level >= 0.25, as measured by ref. 37. Genes' polyA lengths were taken as the mean length measured by ref. 33. Independent maternal genes' degradation rates (for maternal-only genes) were calculated from RNA sequencing measured by refs. 8,14,36,51−53. Only measurements until 6 hpf were used for the calculation.

## Colorimetric RNA in situ hybridization

Fragments of the genes *aif1l*, *cd9b*, and *cth1* were amplified and cloned into pSC-A using the Stratuclone PCR Cloning Kit (Agilent) according to manufacturer instructions, but half-size reactions using the following primers:

*aif1l* (plasmid JFP705): F 5′−gctggaggaaatcaataaggagt−3′ and R 5′−agttcgcttggattaaaacatga−3′

*cd9b* (plasmid JFP706): F 5′−tgaaaagctgggaaaggtacaat−3′ and R 5′−gttacaaaagaatgccagaaagc−3′

*cth1* (plasmid JFP707): F 5′− gtgaagaaatctttccgggtagt−3′ and R 5′−aaaaagcaagcatttgagaattg−3′

They were then linearized using EcoRV (JFP705/*aif1l* and JFP706/*cd9b*, NEB) or NotI (JFP707/*cth1*, NEB) and transcribed using T7 polymerase (JFP705/*aif1l* and JFP706/*cd9b*, Roche) or T3 polymerase (JFP707/*cth1*, Roche) with 10X RNA labeling mix (DIG, Roche) for 3 h. Probes were purified using RNA cleanup columns (Omega E.Z.N.A. Total RNA Kit I), quantified using a Nanodrop, assessed on an agarose gel for successful transcription of a product of the expected size and normalized to 20 ng/μl in HM+ buffer (50% formamide, 5× saline-sodium citrate buffer (SSC), 0.1% Tween-20, citric acid to pH 6.0, 50 μg/ml heparin, 500 μg/ml tRNA), then stored at −20 °C.

Embryos were collected from TL/AB in-crosses, dechorionated, cultured to the appropriate stage in agarose-covered dishes at 28 °C. They were fixed in 4% methanol-free formaldehyde (VWR) at 4 °C overnight. They were rinsed twice for 10 min with PBST (1× PBS + 0.1% Tween-20), 10 min with 50% PBST:50% methanol, rinsed in methanol twice for 10 min, and then permeabilized at −20 °C at least overnight. Embryos were rehydrated (10 min each, 67% methanol:33% PBST, 33% methanol:67% PBST, 3 ×10 min PBST). Embryos were then pre-hybridized in HM+ buffer (50% formamide, 5× SSC buffer, 0.1% Tween-20, citric acid to pH 6.0, 50 μg/ml heparin, 500 μg/ml tRNA) at 70 °C for at least 2 h. Probes were diluted to 1 ng/μl in HM+ buffer and denatured at 70 °C for 10 min. The prehybridization HM+ buffer was replaced by the probe and embryos were incubated with 1 ng/μl probe overnight.

The next morning, the probe was removed and returned to −20 °C for future re-use. Excess probe was removed with first a series of washes that had been prewarmed to 70 °C: 1× 10 min HM buffer (HM+ without heparin and tRNA), 1× 10 min 75% HM:25% 2× SSC, 1× 10 min 50% HM:50% 2× SSC, 1× 10 min 25% HM, 75% 2× SSC, 1× 10 min 0.2× SSC, 2× 30 min 0.2× SSC; then a series of room temperature washes: 1× 5 min 75% 0.2× SSC:25% PBST, 1× 5 min 50% 0.2× SSC:50% PBST, 1× 5 min 25% 0.2× SSC:75% PBST, 1× 5 min PBST. They were blocked for at least 3 h in blocking buffer (2 mg/ml bovine serum albumin, 2% normal donkey serum, Jackson Labs). Finally, they were incubated overnight with anti-digoxigenin antibody coupled to alkaline phosphatase (Anti-Digoxigenin-AP Fab Fragments, Roche 11 093 274 910), diluted 1:5000 in blocking buffer at 4 °C with gentle agitation.

The following morning, the antiserum was removed and discarded, and excess antibody was removed by rinsing embryos 6× 15 min in PBST. They were transferred into staining buffer (100 mM Tris-HCl pH 9.5, 50 mM magnesium chloride, 100 mM sodium chloride, 0.1% Tween-20) by rinsing 3× 5 min. Staining reagent was introduced (225 μg/ml Nitro Blue Tetrazolium and 175 μg/ml BCIP, Roche 11 383 213 001 and 11 383 221 001) and embryos were incubated in the dark, periodically checking their color development under a dissecting

scope until the desired staining had been achieved (15 min–24 h). When the desired staining was achieved, the reaction was stopped by rinsing 3× 5 min. Embryos were dehydrated by passing through a methanol dehydration series, then stored overnight at −20 °C. They were cleared by replacing the methanol with BB/BA (2 parts benzyl benzoate: 1 part benzyl alcohol, Sigma-Aldrich) and imaged on a Zeiss Axioimager Z.1 with a 10× objective (100× total magnification).

### Assaying differential degradation with reporter constructs

**Cloning and transcription of mOxBFP and dendra reporter constructs.** To test whether the untranslated regions of *epcam* mRNA can confer stability in the enveloping layer, we created synthetic mRNA (*dendra* and *mOxBFP*) with either 'neutral' UTRs (standard pCS2 plasmid UTRs−5′ UTR from mouse B-globin gene, very short 3′ UTR from the plasmid backbone) or with the *epcam* UTRs. pCS2-epcam5UTR-mOxBFP-epcam3UTR (JFP709), pCS2-mouseBglobin5UTR-Dendra (JFP682), and pCS2-NLS-mOxBFP (JFP584) were constructed with a mixture of gene synthesis by Twist Bioscience and Gibson cloning. Sequenced verified constructs were linearized prior to the SV40 polyadenylation signal with SnaBI (JFP709), XbaI (JFP682), or XhoI (JFP584), and purified using PCR clean up columns (Omega Cycle Pure Kit). Reporter mRNA was then transcribed using the mMessage SP6 kit (Invitrogen) according to manufacturer's instructions. After DNase treatment, prior to purification, a poly(A) tail was added to the mRNA in vitro using E. coli poly(A) polymerase (ThermoFisher, Poly(A) Tailing Kit) according to manufacturer's instructions, except the amount of poly(A) polymerase was reduced from 4 μL to 1.5 μL per reaction, as suggested by the manufacturer's protocol to produce a poly(A) tail more similar to in vivo length. The mRNA was then purified using RNA cleanup columns (Omega EZNA Total RNA Kit I), and quantified using a Nanodrop. mRNA was run on an agarose gel, and product size was assessed to check for successful transcription before being stored at −80 °C.

**mRNA injection.** Fertilized eggs were collected from TL/AB in-crosses and injected at 1-cell stage with 40 pg each of *dendra* and *mOxBFP* reporter mRNA (with *epcam* UTRs or "neutral" UTRs), in a volume of 1 nL. Injected embryos were cultured at 28 °C until they reached 50% epiboly. Damaged or abnormal embryos were removed. Embryos were fixed in 4% formaldehyde at 4 °C overnight. After fixation, embryos were dehydrated in a gradient from PBST to 1:1 methanol:PBST, and then into methanol for 4 h at −20 °C.

**Hybridization chain reaction (HCR).** *krt8* (which marks the enveloping layer), *mOxBFP* reporter mRNA, and *dendra* mRNA were identified using HCR. HCR probes were designed using the Özpolat lab probe generator, found at https://github.com/rwnull/insitu_probe_generator. The probes were designed with amplifiers B2 and B3. 10 base pairs were skipped from the beginning of the cDNA, and the maximum poly A/T and poly G/C homopolymer length was 5. Probe pairs were ordered in plate format and OPools format (Integrated DNA Technologies), and resuspended in molecular biology grade water to a working concentration of 1 μM. Embryos were rehydrated in a gradient of 1:1 methanol:PBST and then to PBST. Embryos were then pre-hybridized in HCR probe hybridization buffer (Molecular Instruments) for 2 h at 37 °C with shaking at 300 rpm in a ThermoMixer C. To prepare probe working solution, 1 μL of each 1 μM HCR probe was diluted in 500 μL of probe hybridization buffer at 37 °C. Probe hybridization buffer was replaced with probe working solution and hybridized at 37 °C with shaking at 300 rpm overnight (12–16 h). Unbound probe was washed off using HCR probe wash buffer (Molecular Instruments) and pre-amplification was performed with hairpin amplification buffer (Molecular Instruments) for 2 h at room temperature. To prepare the hairpin mixtures, hairpins (3 μM, Molecular Instruments) were pre-

annealed at 98 °C for 90 s and 25 °C for 30 min (ramp rate: −0.1 °C/s). Hairpin mixtures were then diluted to 1:100 in hairpin amplification buffer (Molecular Instruments), and embryos were incubated in the mixtures overnight (12–16 h at 24 °C with 300 rpm shaking). Following amplification, embryos were rinsed at 24 °C in 5× SSCT, followed by washes in PBST.

**Imaging and quantification.** Stained embryos were mounted in a lateral orientation in 1% low-melt agarose gel. Z-stack images of mounted embryos were acquired on a Nikon Eclipse Ti2 inverted resonant scanning confocal microscope with a Nikon DS-R*i*2 camera using a 20×/0.75 NA air objective. Fluorescence of HCR-stained injected embryos was quantified in Fiji (ImageJ). Images were selected for lateral orientation, and yolk was cropped out to minimize the interference of auto-fluorescence on the measurements. The threshold of *krt8* HCR fluorescence was adjusted as necessary to create a mask defined by the enveloping layer. This mask was used to create selections in the *dendra* and *mOxBFP* fluorescence channels. Areas outside the enveloping layer were defined by creating an inverse of the defined mask and de-selecting regions outside of the embryo. Fluorescence of selected regions was quantified using Fiji's "Measure" function. Corrected total fluorescence (CTCF) of a selected region was calculated by subtracting the area of the selected region multiplied by the average background fluorescence from Integrated Density (ID) of the selected region. CTCF values of the *dendra* control signal and *mOxBFP* reporter signal were directly compared.

### Statistics and reproducibility

For all statistical analyses performed in this study, the methods used are addressed in the relevant sections. For single-cell data, only cells with expression of at least 500 genes were retained. For images, embryos were excluded if they were mounted in an inappropriate orientation to visualize the mRNAs of interest. Embryos were randomly collected from natural mating of zebrafish. No statistical method was used to predetermine sample size. The investigators were not blinded to allocation during experiments and outcome assessment.

### Reporting summary

Further information on research design is available in the Nature Portfolio Reporting Summary linked to this article.

## Data availability

Sequencing data generated in this study have been deposited in the NCBI Gene Expression Omnibus, under accessions GSE224113 (bulk RNA-Seq) and GSE224918 (single-cell RNA-Seq) and is freely available. Raw microscopy data for figures presented in this study is available from Zenodo (doi:10.5281/zenodo.10080888). In addition, our portal (https://liorf.shinyapps.io/zebrafish_single_cell_regulation) provides the scientific community with ready access to our data and analysis results. Published datasets used in this study are available in the NCBI Gene Expression Omnibus, under accessions GSE106587 (for expression comparison after 4sUTP injections), GSE52809 (for poly(A) tail lengths), GSE46512 (for ribosome profiling), GSE79213 (for m6A analysis), GSE127780 (for m5C analysis) and GSE79213, GSE84601, GSE148391, GSE120643, GSE32898, GSE56977 (for maternal genes' degradation rates comparisons). Source data are provided with this paper. Published RNA-Seq dataset of sorted germ-cell populations at dome stage used in this study is available in ArrayExpress under accession E-MTAB-8707.

## Code availability

The code used to generate figures and analyses is available from authors upon request.

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

## Acknowledgements
We thank Alex Schier for supporting early aspects of this project, for helpful discussions, and for providing access to sequencing facilities and zebrafish infrastructure. We thank Alex Schier, Eran Meshorer, Yotam Drier, and Sagiv Shifman for critical reading of the manuscript. This research was supported by the Azrieli Faculty Fellowship and European Research Council Horizon 2020 (grant 852451 to MR) and by the National Institutes of Health (K99HD091291 and ZIAHD008997 to JAF).

## Author contributions
M.R., J.A.F., and A.R. conceived the project. M.R. and J.A.F. designed the project, optimized single-cell labeling and collected single-cell data. M.R. and G.N. collected bulk sequencing data. J.A.F. performed single-cell data analyses. M.R. and L.F. performed mRNA dynamics analyses. L.F. and F.E. performed SLAM-Seq analyses. L.F. performed sequence analyses. T.R., A.M., and J.A.F. designed, performed, and analyzed reporter assays and RNA in situ hybridization. M.R. and J.A.F. wrote the manuscript with input from the other authors. All authors read and approved the final manuscript.

## Funding

## Competing interests
A.R. is a founder and equity holder of Celsius Therapeutics, an equity holder in Immunitas Therapeutics and until August 31, 2020 was a SAB member of Syros Pharmaceuticals, Neogene Therapeutics, Asimov and ThermoFisher Scientific. From August 1, 2020, A.R. is an employee of Genentech, and has equity in Roche. All other authors declare no competing interests.
