## [Peer Review File · Nature Communications]

Cell-type-specific mRNA transcription and degradation kinetics in zebrafish embryogenesis from metabolically labeled scRNAseqREVIEWER COMMENTS

Reviewer #1 (Remarks to the Author):

Cell fate changes require the transcription of new mRNAs and the degradation of pre-existing mRNAs. Measuring the balance and dynamics of these two events is challenging, especially at the single-cell level. In the manuscript by Fishman et al., the authors combined metabolic labeling of newly synthesized RNA and scRNA-Seq using zebrafish early embryos as a model system. Using the dense data set and bioinformatical/mathematical approaches, the authors revealed developmental trajectories of early cell fates with quantitative information about maternal and zygotic transcripts. The results were largely consistent with the prevailing views in the field, such as the major contribution of zygotic transcription in most cell type differentiation and the predominant contribution of the maternal transcripts to germline establishment. The single-cell resolution data helped dissect these events in more detail, and contributions and timing of zygotic transcription and maternal mRNA degradation were further categorized into several sub-types. The authors also detected several cis-acting elements related to maternal mRNA stability, which were also consistent with the previous studies.

Although most results are well-expected from previous studies, experimental demonstration of the gene expression kinetics during the maternal-to-zygotic transition at single-cell resolution is a significant advance in the field. In addition, the authors constructed a publicly available database of their data, which will greatly contribute to the communities related to developmental biology, RNA biology, and transcription/chromatin regulation. The bioinformatical and mathematical analysis seemed appropriate, but the assessment of their detail is beyond the expertise of this reviewer; therefore, I would appreciate the evaluation of that part by other expert reviewers. Here I am concentrating my comments on the points related to my expertise listed below.

Major points

The overall consistency of the results of this study with previous publications supports the validity of the approach used in this study. Nevertheless, in this reviewer's opinion, validation experiments are required to strengthen the key new findings obtained by scRNA-Seq analysis. One novel finding of this study is the trajectory-specific stability of maternal mRNAs in the enveloping layer. Therefore, I wonder if time-course *in situ* hybridization can visualize the differential stability of maternal mRNAs. Alternatively, enveloping layer-specific stabilization/destabilization of mRNAs might be recapitulated by injecting reporter mRNAs at the one-cell stage and monitoring reporter expression or mRNA distribution. These experiments were previously applied to germ-cell-specific maternal mRNAs, so it would be important to test if similar validation works for the enveloping layer mRNAs.

Related to the point above, *in situ* hybridization in Figure S3 should ideally be performed by the authors independently of the previous large-scale analysis.

Minor comments

The four groups of genes with differential transcription/degradation kinetics in Figure 5 are another important feature captured by combining the scRNA-Seq and RNA metabolic labeling. More detailed characterization of these four groups regarding gene function would be informative to the readers. Please provide a detailed GO term enrichment analysis with the four groups as a table.

Are there any enveloping layer-specific genes that explain the differential stability of maternal mRNAs in this cell lineage?

Reviewer #2 (Remarks to the Author):

Systematically dissecting the underlying regulation of mRNA transcription and degradation remains a challenge, especially within whole embryos with diverse cellular identities. The manuscript by Fishman et al takes advantage of newer technologies to address this challenge. They collect temporal cellular transcriptomes of zebrafish embryos at three stages, and break them into their newly-transcribed (zygotic) and pre-existing (maternal) mRNA components by combining single-cell RNA-Seq and metabolic labeling. Using this data they identify different regulatory rates between thousands of genes, and sometimes between cell types, that shape spatio-temporal expression patterns. They find that zygotic transcription rather than selective stabilization of RNA is the primary source of cell-type specific expression in early embryos, with germ cells being an exception where stabilization of maternal RNAs is important for restricted expression of a small subset of genes

The idea that distinctions in transcription and degradation will vary for genes across cell types is not particularly surprising, and there are many examples of this phenomena at the individual gene level in the literature. Other ideas raised in this manuscript such as stabilization of RNA as a mechanism for tissue restricted expression specifically in the germline and use of miR430 motifs for rapid degradation of maternal RNAs have similarly been raised in previous publications. However, a strength of this manuscript is that it provides analysis on a larger scale, producing a wealth of new data that will be useful to those interested in early embryogenesis, gene regulation and the maternal to zygotic transition in the process.

In general the manuscript is clearly written, especially given the complexity of some of the expressed ideas/ analysis. My comments related to rigor, analysis and data presentation are mostly minor.

major:

With respect to lines 197-201, it would be helpful to have a discussion of why a gene called as having 65% zygotic transcripts would be considered zygotic only. What's going on with the other 35% of transcripts in these cases if they aren't called zygotic but the authors dont think they are maternal? Some additional clarity in writing / the underlying thought process may be needed.

I'm not quite sure I understand figure 6C-E panels. For each gene, are these binned lineages broken out based on those showing trajectory vs non trajectory dynamics ? (I will admit I struggled a little more in general understanding in this section of the paper.)

minor:

Introduction: Authors write: At the onset of development, embryos are transcriptionally silent and rely on maternally-provided mRNAs and proteins that were deposited in the egg 26. In zebrafish embryos, this maternally controlled period lasts for approximately 3.3 hours, and involves many rounds of cell division. The authors may wish to revise these statements to better reflect the presence of minor ZGA transcripts as early as th 128 cell stage

In figure 1c y axis label overlaps numbers

In panel 1 D it may be more appropriate/less confusing for the y axis to read % labeled RNA (as this small percent of 1 hpf transcripts in ribo depleted could be a technical artifact). One might choose to also label the Y axis in a similar way to be conservative and consistent.

In figure S2D it is unclear what the authors mean by (contamination) in the YSL panel, it doesn't seem to be commented on in the legend or text?

Text associated with panel 2A is somewhat confusing, as it states partitioned genes into 10 equal sized bins, but the graph appears to show 9 bins that are not equally sized? Similarly <5% only seems to encompass the lowest two quintiles in Figure 3A whereas the text says it is three?

In panel 3C given that the authors generally choose to present maternal data on the left and zygotic on the right (which is intuitive from a temporal perspective) they might choose to keep this pattern in this figure panel as well. This reader was temporarily disorientated by the switch to presenting zygotic data on the left.

In lines 240-242, I'm not sure that the author's statement that: "This highlights selective stabilization of maternal mRNAs as the main mechanism for primordial germ cell specification during blastula stages" is appropriate. This is a likely model given the authors data, but they do not provide any actual data linking this selective expression to specification (or the relative contribution to specification). What they show is only a very tantalizing correlation.

It is somewhat confusing to reconcile data in Figure 5B with the text. The authors mention 16/34 genes in group A have cell type restricted expression, but it appears there are more than 34 dots in the A quadrant of panel B. (this reviewer is going on impressions and didn't actually count dots so it is possible I'm wrong on my estimation rather than confused)

Reviewer #3 (Remarks to the Author):

In this study, Fishman and colleagues have combined metabolic RNA labeling with single-cell transcriptomics to analyze the relative contribution of RNA synthesis and degradation to cell-type specific gene expression during zebrafish embryogenesis. They constructed RNA kinetics models to estimate the rates of transcription and degradation within specific cell-types in lineage specification. The authors found that while RNA transcription drives most cell-type specific gene expression, specific maternal transcripts are retained/stabilized in some of the earliest specified cell-types, including germ cells and enveloping layer cells. Further, RNA degradation events are associated with specific sequence motifs. Overall, this study suggests that coordinated regulation of RNA transcription and decay is linked to spatio-temporal patterning of maternal-zygotic gene expression in developing Zebrafish embryos.

This study is experimentally well controlled and executed. The computational analysis is relatively thorough and comprehensively benchmarked. The conclusions are generally supported by the data and analysis. However, I do have a few questions on the study before the study should be accepted for publication.

1. The authors have used GRAND-SLAM to compute the corrected fraction of metabolically labeled RNA (NTR) of each gene in a single cell. Given the sparsity of the scRNA-seq data, how many genes can be reliably analyzed on average for each cell? Also, it is unclear whether all example genes shown in Fig. 2 & 3 are corrected, or only a subset of them are corrected at single-cell levels.
2. The authors have established a kinetics model to calculate the zygotic RNA synthesis rate (α) and maternal RNA degradation rate (γ) along pseudotime, so they can only obtain relative RNA regulatory kinetics rates (termed pseudo-rates). Can they obtain absolute rates if ignoring the cell-type or specific time point? If not, the authors should propose alternative metabolic labeling strategy in the discussion for future studies to enable computation of absolute RNA kinetics rate analysis in a cell-type specific manner.
3. Recent computational analysis tools (e.g. cellDancer, PMID: 37012448) claim that they can also compute relative cell-type specific RNA kinetics parameters only using splicing information (no need for metabolic labeling). It will be informative to apply such novel computational strategy on the current scRNA-seq dataset to verify whether metabolic labeling is still essential to calculate RNA regulatory kinetics parameters.
4. For analyzing the sequence-specific features associated with regulated degradation of maternal RNAs, the authors should also examine codon usage and RNA modifications (if biochemical data available).

Point by point response to reviewers' comments

We thank the reviewers for their thoughtful and constructive feedback. Detailed point-by-point responses are provided below.

Reviewer #1:

Cell fate changes require the transcription of new mRNAs and the degradation of pre-existing mRNAs. Measuring the balance and dynamics of these two events is challenging, especially at the single-cell level. In the manuscript by Fishman et al., the authors combined metabolic labeling of newly synthesized RNA and scRNA-Seq using zebrafish early embryos as a model system. Using the dense data set and bioinformatical/mathematical approaches, the authors revealed developmental trajectories of early cell fates with quantitative information about maternal and zygotic transcripts. The results were largely consistent with the prevailing views in the field, such as the major contribution of zygotic transcription in most cell type differentiation and the predominant contribution of the maternal transcripts to germline establishment. The single-cell resolution data helped dissect these events in more detail, and contributions and timing of zygotic transcription and maternal mRNA degradation were further categorized into several sub-types. The authors also detected several cis-acting elements related to maternal mRNA stability, which were also consistent with the previous studies.

Although most results are well-expected from previous studies, experimental demonstration of the gene expression kinetics during the maternal-to-zygotic transition at single-cell resolution is a significant advance in the field. In addition, the authors constructed a publicly available database of their data, which will greatly contribute to the communities related to developmental biology, RNA biology, and transcription/chromatin regulation. The bioinformatical and mathematical analysis seemed appropriate, but the assessment of their detail is beyond the expertise of this reviewer; therefore, I would appreciate the evaluation of that part by other expert reviewers. Here I am concentrating my comments on the points related to my expertise listed below.

We thank the reviewer for finding our data and analysis is a significant advance in the field, and a useful resource. We have now addressed the questions and comments raised, which considerably strengthened our paper, and we hope the reviewer will find our revised manuscript acceptable for publication.

Major points

The overall consistency of the results of this study with previous publications supports the validity of the approach used in this study. Nevertheless, in this reviewer's opinion, validation experiments are required to strengthen the key new findings obtained by scRNA-Seq analysis. One novel finding of this study is the trajectory-specific stability of maternal mRNAs in the enveloping layer. Therefore, I wonder if time-course in situ hybridization can visualize the differential stability of maternal mRNAs. Alternatively, enveloping layer-specific

stabilization/destabilization of mRNAs might be recapitulated by injecting reporter mRNAs at the one-cell stage and monitoring reporter expression or mRNA distribution. These experiments were previously applied to germ-cell-specific maternal mRNAs, so it would be important to test if similar validation works for the enveloping layer mRNAs.

We agree with the reviewer that an independent validation for the observed effect of maternal mRNA stabilization in the enveloping layer would greatly strengthen this key new finding obtained by our scRNA-Seq analysis. We provide below details on 3 different approaches we took to obtain additional validations for our results.

First, we compared our predictions to similar observations from published literature and performed additional analyses. In our manuscript, we analyzed enveloping-layer maternal mRNA stabilization by a cell-type specific enrichment of old maternal mRNA (Figure 3C), and identified 11 genes (out of 436 genes in the extended list) with a more significant enrichment in enveloping-layer cells of old maternal RNAs than new zygotic RNAs. We now also analyzed predictions by **Holler et. al. (Nat. Comm. 2021)** who observed a similar phenomenon in the enveloping-layer at similar stages. Their results identify 58 genes with a similar stronger enrichment in enveloping-layer cells of old maternal RNAs than new zygotic RNAs (limiting our analysis to genes with a p-value $< 10^{-8}$). In addition, we newly analyzed our normalized pseudo-bulk estimations of gene expression, and found a set of 10 genes with a significant enveloping-layer expression of old maternal RNAs (> 1 normalized pseudo-bulk expression levels) that is higher than their non-enveloping layer maternal expression (normalized pseudo-bulk expression levels by at least 0.5, at both 30% and 50% epiboly stages). Comparing between these predictions, we identified significant overlaps, including 5 genes that are predicted by all 3 approaches, and 13 genes that are predicted by at least 2 of the 3 approaches (**Figure R1a**). We focused on a set of 15 genes which were predicted by at least one of our two analyses (**Figure R1b**), and performed a motif search in their 3'UTRs. We identified (**Figure R1c**) a very strong enrichment for a uridine rich motif (12/15 genes), and a highly similar AU-rich element (ARE) motif (11/15 genes). A similar analysis, using 3'UTR sequences of other (non-stabilized) enveloping-layer genes as background, identified very similar motifs as well (data not shown). These observations provide further validations from additional independent datasets and new analyses, to support a selective enrichment of maternal mRNAs in enveloping-layer cells, and suggest selective stabilization could be involved.

Second, as the reviewer suggested, we attempted to validate the maternal mRNA stabilization of a top candidate (gclm) by HCR in-situ hybridization (**Figure R2**). We attempted to use both exonic and intronic HCR probes, in order to distinguish old (maternal) expression at early stages and new (zygotic) expression. We could successfully capture gclm expression at later stages, where its levels are higher, by exon probe but not by intron probe (**Figure R2a**), possibly due to lower expression levels of introns. Unfortunately, we could not reliably detect gclm expression at early stages (50% epiboly), limiting our ability for validation (**Figure R2b-c**).

Finally, we tried a reporter mRNA approach, inspired by the germ-cell reporter mRNAs, as mentioned by the reviewer. We in vitro transcribed mRNAs with no SV40 polyadenylation signal and added polyA tails using polyA polymerase to produce a consistent length polyA

tail. Reporter mRNAs were Dendra with the 'standard' UTRs from pCS2 (the most common plasmid used for making zebrafish mRNA, which contains a short mouse B-globin 5'UTR and a short 3' UTR) and mOxBFP with either the same 'standard' UTRs from pCS2 or the *epcam* 5' UTR and 3' UTR. Dendra and one of the BFP mRNAs were co-injected, and then HCR was performed to detect the 'control' Dendra mRNA, the 'test' BFP mRNA, and *krt8* (which marks the enveloping layer), and fluorescence was quantitated. While we observed that Dendra and BFP have inherently some difference in their stability or detection in the enveloping layer and deep layer, BFP with 'standard' UTRs behaved much more similarly in the deep layer cells (i.e. the blastoderm) and the enveloping layer cells, while BFP with *epcam* UTRs was much more stable in the enveloping layer compared to the blastoderm. We incorporated these new results into our revised manuscript (Figure 6 and p. 17).

This serves as a proof-of-principle that indeed, some mRNAs exhibit differential stability in enveloping layer cells, and that the UTRs of a gene we identify as being more stable in the enveloping layer than the blastoderm help confer that stability. We hope in the future to continue to optimize this approach (this figure is the result of approximately six rounds of optimization of this experiment to make functional reporter constructs and optimize the imaging to observe them), to test additional genes, and to identify the sequence elements within these UTRs that confer differential stability.

Related to the point above, in situ hybridization in Figure S3 should ideally be performed by the authors independently of the previous large-scale analysis.

We performed in situ hybridizations using a colorimetric approach for the genes *aif1l*, *cth1*, and *cd9b*, as previously presented in this figure. We updated Figure S3 to include our data that all show conclusively that these three genes are maternally provided (ubiquitous at 2.5/3 hpf, prior to the major wave of zygotic transcription), that they later have cell-type specific expression patterns, but that the time at which that cell-type specific expression pattern becomes apparent depends on the rate of maternal mRNA degradation and can be earlier (60% or 75% epiboly for *cth1* and *cd9b*, which have higher maternal turnover rates) or later (not at 60% epiboly, but visible by somitogenesis stages for *aif1l*, which has a slow maternal turnover rate).

Minor comments

The four groups of genes with differential transcription/degradation kinetics in Figure 5 are another important feature captured by combining the scRNA-Seq and RNA metabolic labeling. More detailed characterization of these four groups regarding gene function would be informative to the readers. Please provide a detailed GO term enrichment analysis with the four groups as a table.

We added a new Table S3 with detailed functional enrichment analysis of the 4 groups in Figure 5.

Are there any enveloping layer-specific genes that explain the differential stability of maternal mRNAs in this cell lineage?

This is indeed an intriguing question, and we have performed several analyses aiming to address it.

In the original version of our manuscript, we analyzed all annotated RNA and DNA binding genes for differential expression in the enveloping-layer compared to other cell-types. We identified a subset of 1,192 of these genes with a significant normalized pseudo-bulk expression (normalized row mean across 6 trajectories > -3.5). Clustering (**Figure R3a**) of their mean-normalized expression (each gene to its average across 6 trajectories) identified a cluster of 19 genes with a significant enrichment in enveloping-layer trajectory (**Figure R3b**). As we explain in the manuscript, AU-rich elements were over-represented in 3'UTRs of stabilized maternal transcripts within the enveloping-layer. Therefore, we highlighted in the discussion one interesting candidate, *rbm24a*, which has been linked to mRNA stabilization via AU-rich elements in zebrafish (**Shao et. al., PNAS, 2020**), and exhibit enveloping layer-specific expression during blastula stages. Typically, AU-rich elements destabilize mRNAs, raising the possibility that this process is somehow attenuated within the enveloping-layer. Other genes from this list could also serve as candidates for enveloping-layer specific stability of maternal mRNAs.

To corroborate these results, we newly performed an additional analysis of single-cell rather than pseudo-bulk gene expression. Specifically, we used a Kolmogorov-Smirnov test to compare the distribution of single-cell expression values of 2,102 annotated RNA and DNA binding genes in enveloping-layer and non-enveloping-layer cells, and calculated an effect size by the mean fold change (**Figure R3c**). This analysis identified 13 genes up-regulated in enveloping-layer cells at 30% epiboly and 19 genes up-regulated at 50% epiboly.

Integrating the three analyses, we identified a subset of 10 overlapping genes (**Figure R3c**). Two of these genes, *ptbp1a* and *esrp1*, are specifically annotated as RNA (rather than DNA) binding proteins. In particular, the binding preferences of polypyrimidine tract binding protein 1a (*ptbp1a*) match those of our analysis of 3'UTR elements in maternally enriched enveloping layer genes (**Figure R1**). Moreover, reports show that human PTBP1 enhance the stabilization of mRNAs by binding to ARE sequence elements in 3'UTR (**Ge et. al., Elife, 2016; Fritz etl.al., JBC, 2020; Cho et. al., Sci. Rep., 2019**). Therefore, one or several of these candidates could be involved in enveloping-layer specific stability of maternal mRNAs. We adapted the discussion accordingly (p. 21).

Reviewer #2:

Systematically dissecting the underlying regulation of mRNA transcription and degradation remains a challenge, especially within whole embryos with diverse cellular identities. The manuscript by Fishman et al takes advantage of newer technologies to address this challenge. They collect temporal cellular transcriptomes of zebrafish embryos at three stages, and break them into their newly-transcribed (zygotic) and pre-existing (maternal) mRNA components by combining single-cell RNA-Seq and metabolic labeling. Using this data they identify different regulatory rates between thousands of genes, and sometimes between cell types, that shape spatio-temporal expression patterns. They find that zygotic transcription rather than selective stabilization of RNA is the primary source of cell-type specific expression in early embryos, with germ cells being an exception where stabilization of maternal RNAs is important for restricted expression of a small subset of genes.

The idea that distinctions in transcription and degradation will vary for genes across cell types is not particularly surprising, and there are many examples of this phenomena at the individual gene level in the literature. Other ideas raised in this manuscript such as stabilization of RNA as a mechanism for tissue restricted expression specifically in the germline and use of miR430 motifs for rapid degradation of maternal RNAs have similarly been raised in previous publications. However, a strength of this manuscript is that it provides analysis on a larger scale, producing a wealth of new data that will be useful to those interested in early embryogenesis, gene regulation and the maternal to zygotic transition in the process.

In general the manuscript is clearly written, especially given the complexity of some of the expressed ideas/ analysis. My comments related to rigor, analysis and data presentation are mostly minor.

We thank the reviewer for his/her comments and for finding our paper well-written and a useful resource. We have now addressed the questions and comments raised, and revised our manuscript to better convey several unclear topics and to more accurately state several points. We hope the reviewer will find the revised version of our manuscript is ready for publication.

major:

With respect to lines 197-201, it would be helpful to have a discussion of why a gene called as having 65% zygotic transcripts would be considered zygotic only. What's going on with the other 35% of transcripts in these cases if they aren't called zygotic but the authors don't think they are maternal? Some additional clarity in writing / the underlying thought process may be needed.

The reason we chose to use a 65% threshold for zygotic transcripts is that statistical inference of % newly-transcribed mRNA from T-to-C conversions has more limited accuracy within single cells with fewer reads. Concentrations of 4sU that are tolerated by cells only replace a relatively low number of uridines by 4sU (at most 1 in 10 uridines in our samples, **Figure S1E**). Thus, statistical inference is required in order to estimate fraction of labeled RNA based on the detection of T-to-C conversions. As we show, this inference is more limited in single cells with fewer reads than in bulk analysis, leading in some cases to under-

estimation of the fraction of labeled (newly-transcribed) mRNA. As we show, (**Figure S1F**) within single-cells we estimate labeling of injected controls at < 0.8%, of known maternal genes at < 3.5%, and of known newly-transcribed zygotic genes > 80% (percent is averaged across all cells). The accuracy of statistical inference to estimated zygotic mRNA fractions was further improved by applying GRAND-SLAM to aggregated pseudo-bulk samples (**Figure S1G**), reaching nearly 100% for known zygotic genes (**Figure S1H**).

To accommodate for this estimation accuracy, we used a more conservative threshold of 65% when analyzing single-cell data. However, for some genes, this lower estimation could represent a minimal maternal contribution. We have adjusted our statement in p. 8 to reflect this reasoning, and explain it more clearly in the legend of Figure 3.

I'm not quite sure I understand figure 6C-E panels. For each gene, are these binned lineages broken out based on those showing trajectory vs non trajectory dynamics? (I will admit I struggled a little more in general understanding in this section of the paper.)

Indeed, in figure 6C-E we analyze pseud-bulk expression values based on binning by pseudo-time of either the cells that are within a specific trajectory or the cells that are NOT within a specific trajectory. We added additional explanation to the legend of Figure 6, to reflect the analysis more clearly.

minor:

Introduction: Authors write: At the onset of development, embryos are transcriptionally silent and rely on maternally-provided mRNAs and proteins that were deposited in the egg 26. In zebrafish embryos, this maternally controlled period lasts for approximately 3.3 hours, and involves many rounds of cell division. The authors may wish to revise these statements to better reflect the presence of minor ZGA transcripts as early as the 128 cell stage

We revised the introduction (p. 4) to reflect the presence of a minor ZGA starting at 2.3 hours (128-cell stage).

In figure 1c y axis label overlaps numbers

We corrected the axis label in Figure 1C.

In panel 1 D it may be more appropriate/less confusing for the y axis to read % labeled RNA (as this small percent of 1 hpf transcripts in ribo depleted could be a technical artifact). One might choose to also label the Y axis in a similar way to be conservative and consistent. We agree and corrected the axis label as the reviewer suggest in Figure 1C and 1D, as well as in Figure S1F and S1H.

In figure S2D it is unclear what the authors mean by (contamination) in the YSL panel, it doesn't seem to be commented on in the legend or text?

We do not expect to detect YSL (yolk syncytial layer) in single-cell data, since this syncytial layer is not composed of single cells. Therefore, we refer to it as a contamination, which we assume originates from YSL nuclei. This annotation is explained in the legend of Figure 2, where it first occurs. For clarity, we now also included it in legend of Figure S2A.

Text associated with panel 2A is somewhat confusing, as it states partitioned genes into 10

equal sized bins, but the graph appears to show 9 bins that are not equally sized? Similarly <5% only seems to encompass the lowest two quintiles in Figure 3A whereas the text says it is three?

Our analysis is based on partitioning all genes into 10 equally sized bins based on their % zygotic RNA. However, the two lowest bins had similar values of 0% zygotic RNA, and therefore we merged these two bins, and present them as a single bin. Thus, we have 9 bins, where the first bin is two times larger than the other 8 bins. Since <5% encompass this double-sized bin and the subsequent bin, we refer to it as 3 bins. We modified the legend of Figure 3A, to hopefully better convey this detail of our analysis.

In panel 3C given that the authors generally choose to present maternal data on the left and zygotic on the right (which is intuitive from a temporal perspective) they might choose to keep this pattern in this figure panel as well. This reader was temporarily disorientated by the switch to presenting zygotic data on the left.

We agree with the reviewer's suggestion, and changed the order of the Z and M column in Figure 3C to conserve the same ordering as in other figures in the manuscript.

In lines 240-242, I'm not sure that the author's statement that: "This highlights selective stabilization of maternal mRNAs as the main mechanism for primordial germ cell specification during blastula stages" is appropriate. This is a likely model given the authors data, but they do not provide any actual data linking this selective expression to specification (or the relative contribution to specification). What they show is only a very tantalizing correlation.

We softened this statement (p. 9), and tie this result more to determining which mRNAs are PGC-specific ("the main mechanism that shapes the primordial germ cell transcriptome during blastula stages") rather than making a functional claim about the role of those mRNAs in the PGCs.

It is somewhat confusing to reconcile data in Figure 5B with the text. The authors mention 16/34 genes in group A have cell type restricted expression, but it appears there are more than 34 dots in the A quadrant of panel B. (this reviewer is going on impressions and didn't actually count dots so it is possible I'm wrong on my estimation rather than confused) The counting of 16/34 genes does not refer to the entire set of group A. Rather, it refers to a subset of 16 genes in group A (Figure 5B) that also appear in the list of 34 maternal-zygotic genes with a cell-type restricted expression based on our previous analysis in Figure 3C. This counting shows (by hypergeometric p-value) that cell-type restricted maternal-zygotic genes are more likely (=enriched) to be in group A rather than the other 3 groups in Figure 5B. We adjusted the relevant text (p. 15) to more clearly reflect our counting.

Reviewer #3:

In this study, Fishman and colleagues have combined metabolic RNA labeling with single-cell transcriptomics to analyze the relative contribution of RNA synthesis and degradation to cell-type specific gene expression during zebrafish embryogenesis. They constructed RNA kinetics models to estimate the rates of transcription and degradation within specific cell-types in lineage specification. The authors found that while RNA transcription drives most cell-type specific gene expression, specific maternal transcripts are retained/stabilized in some of the earliest specified cell-types, including germ cells and enveloping layer cells. Further, RNA degradation events are associated with specific sequence motifs. Overall, this study suggests that coordinated regulation of RNA transcription and decay is linked to spatio-temporal patterning of maternal-zygotic gene expression in developing Zebrafish embryos.

This study is experimentally well controlled and executed. The computational analysis is relatively thorough and comprehensively benchmarked. The conclusions are generally supported by the data and analysis. However, I do have a few questions on the study before the study should be accepted for publication.

We thank the reviewer for finding our data and analysis thorough and comprehensive. We have now addressed the comments raised, and revised our analysis based on the reviewer's suggestions. We hope the reviewer will find the revised version of our manuscript can be accepted for publication.

1. The authors have used GRAND-SLAM to compute the corrected fraction of metabolically labeled RNA (NTR) of each gene in a single cell. Given the sparsity of the scRNA-seq data, how many genes can be reliably analyzed on average for each cell? Also, it is unclear whether all example genes shown in Fig. 2 & 3 are corrected, or only a subset of them are corrected at single-cell levels.

Our data measures 1,386 genes per cell on average (with 1 or more UMIs). For our single-cell estimates of gene specific NTR, we analyzed genes which had 3 or more UMIs per cell, and a total of 50 or more UMIs across all cells, resulting in analysis of 195 genes per cell on average. For analysis of per-cell NTR and for pseudo-bulk estimates we considered all UMIs measured in a cell. These numbers are now clearly stated in the methods section (p. 26 and 28) of the revised manuscript.

The gene expression tSNE plots shown in Figures 2 and 3 have all been produced by the same procedure, as we describe in detail in the methods section. Briefly, we estimate a distance matrix based on the pairwise similarities between the cells' embeddings, and use it to calculate tSNE colors for each of the total-RNA, maternal-RNA or zygotic-RNA normalized single cell expression matrices. In addition, the color-scale was normalized per gene, by taking a minimum value at the 30th percentile of maternal-RNA and zygotic-RNA density values, and a maximum value by the maximal total-RNA density value. This color-scale was used for total-RNA, maternal-RNA and zygotic-RNA plots. We now clearly state in legends of Figures 2, 3 and S7 that the color-scale is calculated per gene.

2. The authors have established a kinetics model to calculate the zygotic RNA synthesis

rate (α) and maternal RNA degradation rate (γ) along pseudotime, so they can only obtain relative RNA regulatory kinetics rates (termed pseudo-rates). Can they obtain absolute rates if ignoring the cell-type or specific time point? If not, the authors should propose alternative metabolic labeling strategy in the discussion for future studies to enable computation of absolute RNA kinetics rate analysis in a cell-type specific manner.

In order to dissect absolute degradation rates from metabolic labeling data, the estimation must rely on absolute time rather than pseudotime. This can be achieved by collecting multiple temporal samples at high resolution and analyzing them using scRNA-Seq. Given the (still) high costs of scRNA-Seq sample preparation and sequencing, such an approach would therefore require substantial costs.

However, considering the reviewer's question, we decided to test if we can use absolute decay rates estimated for maternal genes in bulk RNA-Seq data in order to extrapolate absolute rates from our estimated pseudo-rates. We estimated decay rates of maternally provided genes from 6 different published ribosomal depleted RNA-Seq temporal datasets of early zebrafish development (Zhao et. al., 2016, Meyer et. al., 2018 and Medina et. al., 2021, Pauli et. al., 2012 and Yong et. al., 2019). We selected only maternal genes without any evidence for cell-type specific expression in our data, and compared predicted pseudo-rates from our scRNA-Seq data to absolute rates predicted in these datasets (**Figure R4a**). We find a significant correlation (Pearson R between 0.47 and 0.63, depending on dataset), suggesting that our pseudo-rates indeed directly associate to absolute rates. In order to obtain absolute rates from pseudo-rates, we used linear regression between absolute and pseudo-rates. We find that pseudo-rates over estimate absolute rates by 2-fold, and are shifted by an additive factor of 0.2 (1/hr) (**Figure R4b**). Corrected pseudo-rates could reflect more reliable absolute estimates of mRNA degradation rates in our data (**Figure R4c**). We address this approach in the discussion (p. 22) and with an additional supplementary Figure S9.

3. Recent computational analysis tools (e.g. cellDancer, PMID: 37012448) claim that they can also compute relative cell-type specific RNA kinetics parameters only using splicing information (no need for metabolic labeling). It will be informative to apply such novel computational strategy on the current scRNA-seq dataset to verify whether metabolic labeling is still essential to calculate RNA regulatory kinetics parameters.

Indeed, splicing information was used in several publications to analyze mRNA kinetics without a requirement for metabolic labeling. Such approaches have been implemented both in bulk RNA-Seq (e.g., INSPEcT dePretis et. al., 2015) and in single-cell RNA-Seq to infer RNA velocities (La Manno et. al., 2018). While RNA-velocity based approaches robustly estimate cell-state transitions in scRNA-Seq datasets, their per-gene inference could be significantly impacted by biased intron capture within particular genes. First, intron capture by 3' end sequencing depends on the location of introns relative to the transcript ends, which is different between genes. Second, as introns are frequently excised before a polyA tail is added to the mRNA, reverse transcription with oligo-dT could bias intron capture due to differences in splicing rates between introns (e.g., Rabani et. al., Cell 2014). Of course, metabolic labeling approaches could also be impacted by biased incorporation of nucleotides, depending on mRNA sequence and length. However, such biases between genes are generally less significant.

To further investigate the usefulness of metabolic labeling compared to splicing information based on the reviewer's suggestion, we applied cellDancer (Li et. al., 2023) to our data, aiming to infer cell-type specific RNA kinetics from splicing information. Indeed, cellDancer inferred cell state transitions generally matched the expected developmental progression (**Figure R5a**). RNA velocities of some marker genes captured the expected increase during development and showed mono-kinetics phase portraits (e.g., *krt8*, *gsc*, *cdx4*, **Figure R5b**). However, velocities of other genes failed to capture expected expression patterns and show pattern-less phase portraits (e.g., *eve1*, *tbxta*, **Figure R5b**), possibly due to lower intron levels that limit estimation. Importantly, some maternal genes whose levels are expected to decrease over developmental progression show an inverse velocity plot, where early unspecified blastomeres are at the edge of the velocity plot (e.g., *h1m*, **Figure R5b**). Looking at the predicted rates, we find that predicted spliced and unspliced reads in *krt8* indeed accurately captured its specific transcription in the enveloping-layer trajectory (**Figure R5c**). However, predictions for *h1m*, a maternal gene without any expression of its introns, still predict intron expression in early blastomeres, and consequently inaccurately estimate its transcription and degradation rates (**Figure R5c**). When comparing the cellDancer predicted transcription and degradation rates to our estimates (**Figure R5d**), we find a positive correlation in predicted transcription rates, but no correlation in predicted degradation rates.

This analysis suggests that RNA velocity models which rely on intronic reads, are not well suited for analyzing the unique regulation of the maternal-to-zygotic transition. Specifically, maternal genes are not transcribed during early development, but models do predict some residual unspliced reads, which lead to biased estimation. For maternal-zygotic genes, pre-existing maternal copies and newly transcribed zygotic copies could be differently regulated although present in the same cells (e.g., due to differences in modifications and polyA tails). These regulatory features are not captured by standard RNA velocity models. Moreover, gene specific changes to polyA tails (by cytoplasmic polyadenylation and/or deadenylation) are highly frequent in early embryos (e.g., Subtleny et. al., Nature, 2014; Chang et. al., Mol. Cell., 2018), and could further enhance such biases.

4. For analyzing the sequence-specific features associated with regulated degradation of maternal RNAs, the authors should also examine codon usage and RNA modifications (if biochemical data available).

We completely agree with the reviewer that codon usage and RNA modifications are two additional factors which could significantly affect regulation of maternal mRNA degradation, as was previously shown. Therefore, we expanded our analysis to compare our predictions of mRNA stability to these and additional related factors which could affect maternal mRNA degradation.

To analyze the connection between mRNA stability and translation, we first used iCodon (Diez et. al., 2022) to calculate a "codon optimality index" for zebrafish mRNAs. The predicted codon optimality increased as half-life increases (**Figure R6a**), and the difference between codon optimality values predicted at the top or bottom 10% with the rest of the population was highly significant (**Figure R6b**). Second, we analyzed translation efficiency of maternal mRNAs that were calculated from ribosomal foot-printing data (Chew et. al., 2013). We find a strong enrichment of high translation efficiencies in mRNAs with 10% longest half-lives, starting at 1 hpf (**Figure R6c**), and throughout many other developmental

stages. The enrichment of low translation efficiencies in mRNAs with 10% shortest half-lives was weaker, but significant at 4 hpf ($p < 2.6e-6$). Finally, we also analyzed the 5'UTR sequences of mRNAs, which could affect their translation as well (**Figure R6d**). Our results show that an AUG triplet (**Figure R6e**) is significantly associated with a lower mRNA stability. It was shown that the occurrence of upstream ORFs (uORFs) in the 5'UTR reduces the translation efficiency in the main ORF (Chew et. al., 2013), and thus could explain the observed association with lower mRNA stability.

To investigate the effect of mRNA modifications, we analyzed two biochemical datasets, that measured m6A modification (Zhao et.al., 2017) and m5C modification (Liu et. al 2022) of mRNAs in zebrafish embryos. Previously, it was reported that clearance of m6A-modified maternal mRNAs is facilitated by the m6A-binding protein Ythdf2, and its loss delayed their degradation (Zhao et.al., 2017). On the other hand, the m5C modification was reported to delay maternal mRNA degradation (Yang et. al 2019). Our results indeed associated both modifications in maternal mRNAs with differences in their half-lives. Interestingly, the effect of both modifications seems to change during development. While m6A modifications (**Figure R7a**) at early times (0-4 hpf) are associated with shorter half-life (faster decay), as previously reported, m6A modifications at later timepoints (6-8 hpf) are in fact associated with longer half-life (slower decay, **Figure R7b**). Similarly, the occurrence of m5C modification was significantly associated with longer half-lives (slower decay) at later timepoints (6-8 hpf) as reported (**Figure R7c**), but also with shorter half-life (faster decay) at earlier timepoints. These results suggest a similar effect to both modifications, where an early modification is associated with faster decay, but a late modification is associated with a slower decay of maternal transcripts.

We thank the reviewer for this suggestion, that allowed us to expand our sequence-based analysis, and incorporated these additional results in our revised manuscript (p. 18) and Figure S7.

Figure R1

Figure R1. (A) Comparison of maternally stabilized genes in enveloping-layer cells by 3 different approaches. Blue (Junker): 58 genes maternally enriched in enveloping-layer by Holler et. al. Nat. Com. 2018. We selected genes with $p < 1e-7$ and stronger enrichment of the unlabeled RNA (maternal). Orange (expression): 11 genes maternally enriched in enveloping-layer by our cell-type enrichment analysis, and stronger enrichment of the maternal RNA. Green (temporal): 10 genes maternally enriched in enveloping-layer by comparing pseudo-bulk expression between enveloping-layer trajectory and not-trajectory cells. We selected genes with maternal expression > 1 and a difference > 0.5 in both 30% and 50% epiboly samples. Numbers of the overlap between each group are indicated. **(B)** A table showing the 15 genes present in either expression (orange) or temporal (green) groups (green: present, red: missing). For genes not in the temporal (green) group also an explanation why. 3'UTR length according to sequences downloaded from ensembl biomart, and the number of appearances of two AU-rich kmers (UUAUUU, AUUUA). **(C)** Results of motif enrichment analysis using STREME (Bailey, 2021) with default parameters and --rna parameter, for the set of 15 genes present in either the expression (orange) group or temporal (green) groups, using the longest 3'UTR for each gene. Background was 2-order shuffle of input sequences.

Figure R2

Figure R2. HCR staining for *gclm* mRNA in zebrafish embryos. Red: probes designed to match *gclm* introns. Green: probes designed to match *gclm* exon. Blue: nuclear stain. **(A)** Staining at later developmental stages (24 hpf), when *gclm* is expressed in the lens and kidney (pronephros). We can indeed see our exon probes stain the expected structures, thus validating their correct targeting. However, we do not get any signal from intronic probes. As a result, we cannot use intronic probes to make any claims about whether any earlier staining would be maternal or zygotic mRNA. **(B)** Staining at earlier developmental stages (5 hpf) showing expression of *gclm* by exon and intron probes. Signal by both staining is very low. **(C)** High-resolution image of staining at earlier developmental stages (5 hpf) showing expression of *gclm* by exon probe. Due to low expression, its localization to the enveloping-layer is had to determine based on stains.

Figure R3

Figure R3. (A) Total pseudo-bulk expression across all samples in each of the 6 trajectories (columns) for genes (rows) annotated as RNA binding proteins and expressed > -3.5 normalized RNA levels. Values for each gene (row) were normalized by its mean across all cell types (blue: low relative expression; red: high relative expression). Hierarchical clustering was performed using ComplexHeatmap, and divided to 3 clusters. Cluster #3 (green box) includes genes enriched in the enveloping-layer. (B) Zoomed-in view of cluster #3 (green: relative expression). Red: genes with an RNA binding term in zfin.org. (C) Volcano plots for testing the enrichment of genes in the enveloping layer. y-axis: one-sided Kolomogorov-Smirnoff p-values between RNA expression in enveloping-layer cells and all other cells. x-axis: effect size (mean fold-change between groups). Light pink: genes with an RNA or DNA binding annotation; dark pink: genes with an RNA binding GO term. (D) Intersection between the genes in (B) (blue) and significant genes in (C): 30% epiboly (green), 50% epiboly (orange).

Figure R4

Figure R4 (A) Correlation between degradation rates calculated for maternal genes in different datasets (y-axis; name of dataset indicated on top of each plot) and pseudo degradation rates from current analysis (x-axis) (colorscale indicates density; yellow: denser, purple: sparser). Number of genes (N) and Pearson correlation (R) indicated in top left of each plot. **(B)** Correlation between degradation rates calculated for maternal genes in Medina et al., 2021 (ribo) dataset, and pseudo degradation rates from current analysis (x-axis) (colorscale indicates density; yellow: denser, purple: sparser). Black line represents a linear regression model to connect the two estimates. Model parameters (red) indicated on top. **(C)** Frequency (y-axis; % of genes) of degradation rate (left) and half life (right) of the original pseudo-rates (orange) and rates after correction using the linear model from (B) (blue).

Figure R5

Figure R5. (A) UMAP projection of 8,226 single cells from zebrafish embryos colored by 13 distinct cell-type clusters. Arrows show the velocities derived from cellDancer. (B) The velocities derived from cellDancer for several zygotic cell-type marker genes and maternal genes illustrated on a phase portraits. The cells are colored according to the cell types. (C) Several parameters estimated by cellDancer for two genes (krt8 - top, h1m - bottom) color-coded on UMAP projection of cells. Specific feature is indicated on top of each plot. (D) Pearson correlation of transcription (left) and degradation (right) rates that were estimated by cellDancer (x-axis, log₂) and by our analysis (y-axis, log₂). Pearson R is noted on top.

Figure R6

Figure R6. (A) Distribution of codon optimality predictions (y-axis) in each of the 10 half-life quantiles (x-axis). The red line connects the medians of all quantiles. To calculate codon optimality, CDS sequences were downloaded from ensemble biomaRt and filtered to include only sequences starting with a start codon, ending with a stop codon, and sequence lengths being multiples of three. Codon optimality was calculated using the icodon R package (Diez et al., 2022), keeping only the highest value per gene to keep only a single value per gene. (B) Frequency (y-axis) of codon optimality predictions. For 10% of genes with the longest pseudo-half-life (purple, solid) or other (purple, dashed), and for 10% of genes with shortest pseudo-half-life (green, solid) or other (green, dashed). P-value of a Wilcoxon rank sum test is indicated. (C) Frequency (y-axis) of mean of translation efficiency (x-axis) as measured by (Chew et al. 2013) at different developmental stages. Colors as in (B). P-value of a one-sided Kolmogorov-Smirnov test (FDR < 1%) is indicated. (D) Volcano plot for the enrichments of short sequences of length 3-8 bases (k-mers) based on difference between maternal pseudo-half-life parameter (\log_2) for genes that include this short sequence in their longest annotated 5'UTR sequence and those that do not. Plots show the significance (y-axis, $-\log_{10}(p\text{-value})$, Kolmogorov-Smirnov FDR < 1%) and the effect-size (x-axis, difference in the standard mean of each of the two distributions). Horizontal dashed line is 1% FDR, vertical dashed line is an absolute effect size of 0.13. (E) Distribution (y-axis; % of genes) of half-lives (\log_2 , hours, x-axis) for genes with an AUG in their 5'UTR sequence (blue) or those without (gray).

Figure R7

Figure R7. (A) Distribution of predicted mRNA pseudo-half-lives for genes with evidence for m⁶A modification as predicted by (Zhao et. al., 2017). For each of the timepoints, a gene was considered to contain m⁶A modification in both, if both m⁶A-seq and m⁶A-CLIP-seq showed a modification, in single if a modification was observed in only one dataset, neither if there was no modification in either dataset, or no data if no measurement was performed. P-values represent a Wilcoxon rank sum test of the neither group with each of the other 3 groups. Significance is indicated by *: $p \leq 0.05$, **: $p \leq 0.01$, ***: $p \leq 0.001$, ****: $p \leq 0.0001$. **(B)** Distribution of $\log_2(\text{half-life})$ (y-axis) in genes with m⁶A modifications at different stages. For each gene, we count number of datasets modification was detected (m⁶A-seq or m⁶A-CLIP-seq, at each timepoint). A gene was considered modified if count was > 6 (left) or ≥ 6 (right) for the early timepoints (blue), 4 for the late timepoints (green), both (purple) or neither (red). P-values represent Wilcoxon rank sum test of the none group with each of the other groups. Significance labeled as in (A). **(C)** Distribution of $\log_2(\text{half-life})$ (y-axis) in each timepoint analyzed (x-axis) with an m⁵C modification (green) and without (red). A gene was considered m⁵C modified if it contained a location with an m⁵C level ≥ 0.25 as measured by (Dies et. al., 2022). P-values represent a one-sided Kolomogorov-Smirnoff test between half-life of modified and non-modified genes at each timepoint.

REVIEWERS' COMMENTS

Reviewer #1 (Remarks to the Author):

In the revised manuscript by Fishman et al., the authors addressed the concerns and questions from the reviewers in a satisfactory manner. Especially, mRNA reporter analysis visualizing the EVL-specific mRNA stabilization/translation is impressive. I have no further comments, and support publication of this exciting study.

Reviewer #2 (Remarks to the Author):

All of my concerns regarding this manuscript have been addressed.